# Optimal Algorithms for Learning Partitions with Faulty Oracles

**Adela Frances DePavia**
University of Chicago
adepavia@uchicago.edu

**Olga Medrano Martín del Campo**
University of Chicago
omedranomdelc@uchicago.edu

**Erasmo Tani**
University of Chicago
etani@uchicago.edu

## Abstract

We consider a clustering problem where a learner seeks to partition a finite set by querying a faulty oracle. This models applications where learners crowdsource information from non-expert human workers or conduct noisy experiments to determine group structure. The learner aims to exactly recover a partition by submitting queries of the form "are $u$ and $v$ in the same group?" for any pair of elements $u$ and $v$ in the set. Moreover, because the learner only has access to faulty sources of information, they require an error-tolerant algorithm for this task: i.e. they must fully recover the correct partition, even if up to $\ell$ answers are incorrect, for some error-tolerance parameter $\ell$. We study the question: for any given error-tolerance $\ell$, what is the minimum number of queries needed to learn a finite set partition of $n$ elements into $k$ groups? We design algorithms for this task and prove that they achieve optimal query complexity. To analyze our algorithms, we first highlight a connection between this task and correlation clustering. We then use this connection to build a Rényi-Ulam style analytical framework for this problem, which yields matching lower bounds. Our analysis also reveals an inherent asymmetry between the query complexity necessary to be robust against false negative errors as opposed to false positive errors.

## 1 Introduction

Learning cluster structure from data is a fundamental task in machine learning. While the statistical setting is typically concerned with using batch data to approximately recover cluster structure with high probability, some applications allow for the learner to make explicit queries, and some require exact recovery guarantees.

We highlight two key settings. In several scientific domains, particularly bioinformatics, researchers conduct physical experiments to learn whether two objects are part of the same class [9, 11, 20, 35]. Another major application is learning cluster structure by collecting information from human workers via crowdsourcing services [15, 30, 37]. While some traditional methods focus on querying workers for class labels, alternative approaches use simpler *same-cluster* queries. For example, Vinayak and Hassibi [37] point out that, given images of birds, it may be easier for non-experts to correctly answer the question "Do these two birds belong to the same species?" as opposed to "What species does this bird belong to?" In both of these application domains, the learner typically seeks to minimize the number of queries made, as queries require carrying out potentially expensive measurements or time-consuming experiments.

We consider a very general setting and develop algorithms that make no assumptions about the specific type of data involved, but rather rely solely on the same-cluster queries to infer structure. In particular, we model this task by assuming that the algorithm only interfaces with the data via a *same-cluster oracle*. In this setting, the learner obtains information about a hidden partition $\mathcal{C}$ of a finite set $V$ by repeatedly choosing two elements $u, v \in V$ and asking questions of the form "Are $u$

38th Conference on Neural Information Processing Systems (NeurIPS 2024).

and $v$ part of the same cluster?", i.e. given $\mathcal{C}$ and $V$, a same-cluster oracle is an oracle $\alpha_{\mathcal{C}}$ that takes as input $u$ and $v$ in $V$, and returns $1$ if $u$ and $v$ belong to the same cluster in $\mathcal{C}$, and $-1$ otherwise. The power and limitations of same-cluster oracles have been studied in a variety of settings, including the clustering problem described above [8, 26, 33, 34].

However, previous work on this model makes the arguably unrealistic assumption that all queries return the correct answer. In the motivating applications, queries are often at risk of failure that results from relying on non-expert workers or suffering from experimental noise. The learner's goal is to recover the partition in spite of these errors. Such errors are also often not persistent: in the presence of noise, repeating an experiment multiple times may yield different answers, and querying different human workers may result in conflicting responses. Nonetheless, practitioners may need to achieve exact recovery of the underlying partition. However, existing theory either focuses on the error-free regime [26, 33], assumes that errors are persistent [16, 32], or focuses on probabilistic guarantees rather than exact recovery [15].

To address this literature gap, we study algorithms for partition learning via same-cluster oracles that are robust to non-persistent errors. A typical way to incorporate uncertainty and noise is to assume that errors occur independently at random with some small probability on every query. However, it is not possible to guarantee exact recovery of the underlying clusters in this model. Instead, we focus on a model in which the learner sets an error-tolerance parameter $\ell$, and they require guaranteed full recovery of the hidden partition as long as the error incurred is within this tolerance. In particular, we say a same-cluster oracle $\alpha_{\mathcal{C}}$ is $\ell$-faulty if it may return an incorrect answer up to $\ell$ times. We do not assume that errors are persistent: if a single pair of elements is queried repeatedly, an $\ell$-faulty oracle may return inconsistent responses. We define the $\ell$-*bounded error partition learning* ($\ell$-PL) problem, as the problem of exactly recovering a hidden partition via access to an $\ell$-faulty same-cluster oracle.

We design algorithms for the $\ell$-PL problem and related variants, and analyze their query complexity. We introduce a two-player game based on *correlation clustering*, and we show that the minimax value of this game is closely linked to the query complexity of the $\ell$-PL problem. We then use this game as a framework to give tight lower bounds for the query complexity of this task, proving the proposed algorithms are optimal.

We find that the $\ell$-PL problem occupies a unique position at the intersection of different research streams. In fact, the study of this problem complements work on clustering with same-cluster query advice, and the techniques used for its analysis draw from the theory of graph learning with oracle queries and the study of Rényi-Ulam liar games[1]. We are hopeful that the impact of these connections will go beyond the results presented in this paper.

## 1.1 Background and related results

**Clustering with same-cluster oracles**  In the error-free regime, Liu and Mukherjee [26] studied the query complexity of recovering partitions of a finite set in different oracle models. They prove tight lower bounds on the query complexity of learning partitions with error-free same-cluster oracles, and point out that an algorithm proposed by Reyzin and Srivastava [33] exactly achieves this query complexity.

In many motivating applications of this model queries may return the wrong answer, such as when labels result from non-expert human input or noisy scientific experiments. Vinayak and Hassibi [37] consider multiple query models for collecting information from human workers, pointing out that same-cluster queries can accomplish similar end goals as label queries while potentially being easier questions for non-experts to answer correctly. They also provide an algorithm that works for the setting in which triangle queries–which ask the worker to provide all pairwise relationships between three data points–are made. Mazumdar and Saha [30] initiate the formal study of clustering with same-cluster oracles in the presence of persistent i.i.d. errors. They point out that under these assumptions, learning a partition has a strong relationship to recovering community structure in the stochastic block model (SBM). This work inspired a productive line of research on the i.i.d. noise model, which also leverages connections to the SBM to study related problems. In fact, this problem has been studied in the setting when $k = 2$ [22], when the underlying clusters are nearly-balanced [32], and in the semi-random noise model, in which errors occur with some i.i.d. probability, but when they do occur the (erroneous) answer may be chosen adversarially [16]. Models for non-persistent

---

[1]We provide the reader with relevant background from these areas in Section 1.1 and Appendix A.

error have also been considered. Chen et al. [15] study same-cluster queries in the presence of i.i.d. error, allowing for repeated querying of pairs. They give an efficient algorithm with recovery guarantees that does not require a-priori knowledge of the probability of query failure. Similar results were also subsequently established in a recent paper by Gupta et al. [23]. Our work complements this line of research by considering a setting in which exact recovery is possible in spite of errors and providing a full characterization of the query complexity of the problem in this setting.

**Rényi-Ulam games** In his autobiography, Stanisław Ulam, introduced the following two-player game [36]. One player (the responder) thinks of a number in $x \in [N]$ for some $N \in \mathbb{N}$, and the other player (the questioner), given $N$, tries to guess $x$ by asking only yes/no questions. The main twist to this setup is that the responder is allowed to lie up to $\ell$ times. The term "Rényi-Ulam games" has since been used to identify a wide range of problems involving asking questions to an oracle who is allowed some limited amount of lying (see e.g. [31]). The question of finding the worst-case query complexity of the $\ell$-PL problem can be naturally formulated as a Rényi-Ulam game, in which the learning algorithm takes the role of the questioner, and the oracle plays the part of the responder.

**Correlation clustering** Bansal et al. [10] introduced correlation clustering. In this problem, one is given a signed graph $G = (V, E, \sigma)$, where $\sigma : E \to \{\pm 1\}$ is a function representing one's prior belief about which pairs of vertices belong to the same cluster. The goal of the problem is to return a partition of the vertices of $G$ into clusters that minimizes the amount of disagreement with the edge signs given by $\sigma$. The problem is known to be NP-Hard. Many variations of problem have been proposed, including versions with weighted edges [2, 17], a version where the number of clusters is constrained to some fixed $k$ [19], and an agreement maximization setting. Different assumptions have also been studied, such as instance stability and noisy partial information [27–29].

## 1.2 Discussion of the Model

In the previous sections, we introduce a problem in which a learner has to exactly recover a hidden partition by making same-cluster queries that could be subject to up to $\ell$ errors. Other plausible models for learning partitions with errors could be considered. In this section, we discuss some of the key design features of the model and provide justification and examples.

In general, even in the regime in which $k$ the number of clusters is known to the learner, it is not possible to recover the partition exactly unless one is able to resolve all but at most 2 of the pairwise same-cluster relationship between the elements. This follows from a result of Reyzin and Srivastava (this is a key idea leveraged in the proof of Proposition 3 in [33]). In particular, simple extensions of these core arguments imply that in the setting in which the answers to the queries are persistent, it is impossible to solve the problem even for small constant values of of $\ell$ [18, 33]. Furthermore, this implies that even in the non-persistent error-model, one cannot solve the problem for a value of $\ell$ that grows linearly with $n$. Below, we give two general motivating examples, in the technology and scientific domains respectively, illustrating the role of $\ell$ in partition learning tasks.

**Example 1: Robustness to Misinformation** Consider a setting where a user is trying to cluster a dataset by crowdsourcing information in the form of same-cluster questions. However, the user suspects that an ill-intentioned competitor organization is attempting to corrupt the learning process by entering a number of bad actors in the crowd to strategically mislabel queries. If the user selects a new person every time they submit a query, then the number of adversarial answers they encounter is finite and does not grow with the number of queries submitted. In this scenario, $\ell$ plays the role of a security parameter, and the algorithm is guaranteed to be robust to up to $\ell$-many poisoned responses. The user can set $\ell$ based on, e.g., their prior belief about the resources of the competitor organization. Our results can be interpreted as quantifying the cost (in queries / crowd size) of implementing a fixed security parameter $\ell$.

**Example 2: Trustworthy Science** Consider a setting in which a scientist is attempting to group items into classes by running experiments that reveal whether two items are in the same class, such as the example of clustering compatible molecules described in Gupta et al. [23]. The scientist has limited resources (e.g. limited materials or time) and can only conduct a finite number of experiments. Our results allow the scientist to derive the maximum number of errors to which their learning procedure can be tolerant, given their fixed query budget. They can use this maximum value as

| | QUERY COMPLEXITY | |
| --- | --- | --- |
| | $k$-KNOWN | $k$-UNKNOWN |
| WEIGHTED $\ell$-PL (THM. 1, 3, AND 5) | $\Theta\big(RS^k(n,k) + (n-k)\ell_{\text{YES}} + k^2\ell_{\text{NO}}\big)$ | $O\big(RS^u(n,k) + (n-k)\max\{\ell_{\text{YES}}, \ell_{\text{NO}}\} + k^2\ell_{\text{NO}}\big)$ |
| $\ell$-PL (COR. 2 AND 4, THM. 5) | $\Theta\big(RS^k(n,k) + (n-k)\ell + k^2\ell\big)$ | $\Theta\big(RS^u(n,k) + (n-k)\ell + k^2\ell\big)$ |
| $\ell$-PL$^{\text{FN}}$ (THM. 1, 3, AND 19) | $\Theta\big(RS^k(n,k) + k^2\ell\big)$ | $\Theta\big(RS^u(n,k) + (n-k)\ell + k^2\ell\big)$ |
| $\ell$-PL$^{\text{FP}}$ (THM. 1, 3, AND 21) | $\Theta\big(RS^k(n,k) + (n-k)\ell\big)$ | $\Theta\big(RS^u(n,k) + (n-k)\ell\big)$ |
| ERROR-FREE (RESULTS FROM [26, 33]) | $RS^k(n,k)$ | $RS^u(n,k)$ |

Table 1: The query complexity of the different variants of the $\ell$-PL problem, for both the $k$-known ad the $k$-unknown setting. We obtain matching upper and lower bounds for all the variants with the exception of the weighted $\ell$-PL problem in $k$-unknown setting, for which the upper bound does not match the lower bound (given for the $k$-unknown case) exactly. The results are given in terms of the complexity $RS^k(n,k)$ and $RS^u(n,k)$ (defined in Equations 1 and 2) of solving the problem without errors.

the setting for $\ell$, and then use our algorithms to guide their choice of experiments. Our analysis would then allow them to measure the significance of their findings by quantifying the number of experiments that would need to have failed for the finding to be incorrect.

## 2 Technical preliminaries

**Basic definitions** Given a positive integer $n$ we denote with $[n]$ the set $\{1, ..., n\}$. Given any finite set $V$ we denote with $\binom{V}{2}$ the set $\{\{i,j\} \subseteq V \mid i \neq j\}$. A graph $G = (V, E)$ is a pair containing a finite vertex set $V$ and a subset $E \subseteq \binom{V}{2}$. Given a finite set $V$ a $k$-partition of $V$ is a collection of $k$ pairwise disjoint non-empty subsets $\mathcal{C} = \{C_1, ..., C_k\}$ of $V$ such that $\bigcup_{a=1}^{k} C_a = V$. We will denote by $n$ the cardinality $|V|$ of $V$ and we may assume without loss of generality that $V = [n]$. Given two elements $u$ and $v$ of $V$, we write $u \sim_{\mathcal{C}} v$ (resp. $u \not\sim_{\mathcal{C}} v$) for the statement $\exists C \in \mathcal{C}, \{u,v\} \subseteq C$ (resp. $\nexists C \in \mathcal{C}, \{u,v\} \subseteq C$). We denote by $[u]_{\mathcal{C}}$ the equivalence class of $u$ with respect to $\mathcal{C}$, i.e. $[u]_{\mathcal{C}}$ is the unique $C \in \mathcal{C}$ containing $u$. Given two partitions $\mathcal{C}_1$ and $\mathcal{C}_2$ we say $\mathcal{C}_2$ is a *refinement* of $\mathcal{C}_1$ if for every $C \in \mathcal{C}_2$ there is a $C' \in \mathcal{C}_1$ such that $C \subseteq C'$.

## 3 A hierarchy of problems

In the introduction we consider learning a partition $\mathcal{C}$ of a finite set $V$ by asking questions of the form "Are $u$ and $v$ part of the same cluster in $\mathcal{C}$?", where up to $\ell$ of the answers may be incorrect. We refer to this as the $\ell$-PL problem. We characterize the hardness of the $\ell$-PL problem by considering a family of related tasks. In particular, we introduce a hierarchy of problems with different degrees of difficulty (see Figure 1) and establish matching upper lower bounds in nearly all of these tasks. We summarize our results in Table 1.

At the bottom of the hierarchy, we have the problem first studied by Reyzin and Srivastava [33] of learning partitions with same-cluster queries, where the answer to every query is guaranteed to be

correct. They give an algorithm that can learn the underlying partition with:

$$RS^k(n, k) \stackrel{\text{def}}{=} n(k-1) - \binom{k}{2} \tag{1}$$

many queries when the number $k$ of clusters is known to learner and:

$$RS^u(n, k) \stackrel{\text{def}}{=} nk - \binom{k+1}{2} \tag{2}$$

when $k$ is unknown to the learner. Recently, Liu and Mukherjee [26] showed that no algorithm can guarantee full recovery of the underlying partition in fewer that $RS^k(n, k)$ (resp. $RS^u(n, k)$) queries when $k$ is known (resp. unknown) to the learner, showing that the algorithms of Reyzin and Srivastava are optimal down to the exact constants. Since this is the "easiest" version of the problem we will be considering, the lower bounds of Liu and Mukherjee immediately imply the same lower bounds for all the other problems in the family.

We then consider two variants of this problem, each only admitting one-sided error. In the first variant, which we refer to as the *$\ell$-bounded error partition learning with false positives* ($\ell$-PL$^{\text{FP}}$) problem, the oracle might return the answer $+1$ even when two elements are not part of the same cluster. We refer to this kind of fault as a *false positive* answer. Here, we restrict the oracle to return at most $\ell$ many false positive answers, for some fixed positive integer $\ell$, and to always return $-1$ when the two elements being queried are not part of the same cluster.

In the second variant, which we refer to as the $\ell$-PL$^{\text{FN}}$ problem, the oracle behaves the opposite way, and it may return $-1$ even if the two elements being queried are part of the same cluster in the hidden partition (a *false negative answer*), but it cannot return a false positive answer. Similarly to the previous variant, we restrict the number of false negative answers to be at most some fixed $\ell$. We give precise definitions of the $\ell$-PL$^{\text{FP}}$ and $\ell$-PL$^{\text{FN}}$ problems below.

The $\ell$-PL problem is then a generalization of all of the three variants described above (false positives, false negatives and no error), and hence, it inherits hardness results from all three of these problems. Instead of directly designing an algorithm for the $\ell$-PL problem, we consider a yet more general version of the problem, and use results from that generalized version to obtain algorithms for all of the other problems in the class. This final variant is a more fine-grained problem, which allows false positive answers and false negative answers to incur different penalties. In particular we consider a weighted version of $\ell$-PL, which is formalized in the following definition:

**Definition 1** ( $(\ell_{\text{yes}}, \ell_{\text{no}})$-Faulty Oracle and Weighted $\ell$-PL Problem). Let $V$ be a finite set, and $\mathcal{C}$ be a partition of $V$. A $(\ell_{\text{yes}}, \ell_{\text{no}})$-*faulty oracle* for $\mathcal{C}$ is an algorithm $\alpha_{\mathcal{C}}$ which, given as input a pair of elements $uv$, returns a value $r = \in \{\pm 1\}$ so that for any sequence of queries $\{u_t v_t\}_{t \in [T]}$ the sequence of responses $\{r_t = \alpha_{\mathcal{C}}(u_t v_t)\}_{t \in [T]}$ satisfies:

$$\frac{|\{t : u_t \not\sim_{\mathcal{C}} v_t \wedge r_t = +1\}|}{\ell_{\text{yes}}} + \frac{|\{t : u_t \sim_{\mathcal{C}} v_t \wedge r_t = -1\}|}{\ell_{\text{no}}} \leq 1.$$

Note that we assume the oracle can maintain an internal state, so its answers may depend on the query history. The *weighted* $\ell$-PL problem asks one to recover an unknown $k$-partition of $V$ given access to an $(\ell_{\text{yes}}, \ell_{\text{no}})$-faulty oracle for $\mathcal{C}$.

Observe that by setting $\ell_{\text{yes}} < 1$ (resp. $\ell_{\text{no}} < 1$) one would require the oracle to not return any false positive (resp. false negative) answer. As a convention, we will write $\ell_{\text{yes}} = 0$ or $\ell_{\text{no}} = 0$ to indicate any setting that ensures the oracle may not return any false positive or false negative answers respectively. We note that, up to rescaling $\ell_{\text{yes}}$ and $\ell_{\text{no}}$ by a factor of 2, this problem is equivalent to a version of the problem in which the oracle has separate constraints for the number of false positive answers and false negative answers, a simple fact which we prove in Appendix B.

Given this definition, the (unweighted) $\ell$-PL can be cast as a homogeneous version of the above, one in which $\ell_{\text{yes}} = \ell_{\text{no}}$, as follows:

**Definition 2** ($\ell$-Faulty Oracle and $\ell$-PL Problem). An $\ell$-faulty oracle is an $(\ell, \ell)$-faulty oracle. The $\ell$-PL problem is the problem of learning a partition with access to an $\ell$-faulty oracle.

The $\ell$-PL$^{\text{FP}}$ and $\ell$-PL$^{\text{FN}}$ problems discussed above can now be formally described as special instances of the weighted $\ell$-PL problem.

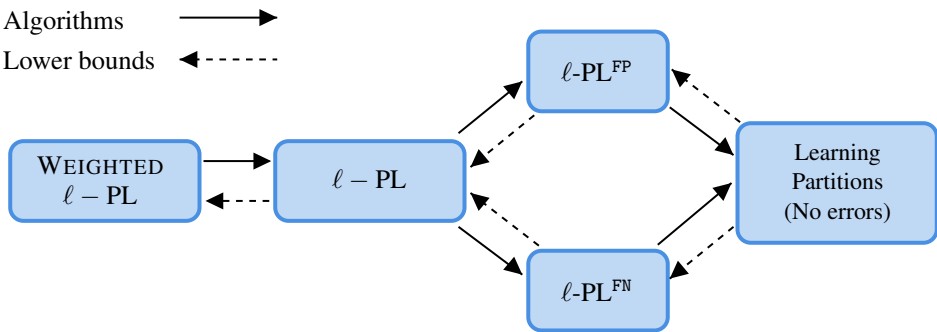

Figure 1: The lattice of problems considered in this paper. Algorithmic results propagate from left to right while lower bound results propagate right to left.

**Definition 3** ($\ell$-PL$^{\text{FP}}$ Problem and $\ell$-PL$^{\text{FN}}$ Problem). The $\ell$-PL$^{\text{FP}}$ problem is the problem of learning a partition with access to an $(\ell, 0)$-faulty oracle. Intuitively, this corresponds to the problem of learning partitions subject to at most $\ell$ false-positive responses and no false negatives. The $\ell$-PL$^{\text{FN}}$ problem is the problem of learning a partition with access to an $(0, \ell)$-faulty oracle. This corresponds to learning partitions subject to at most $\ell$ false-negatives responses and no false positives.

In the rest of the paper, we will always assume that the set $V$, its cardinality $n$, and the parameters $(\ell_{\text{yes}}, \ell_{\text{no}})$ are known to the learner. We will consider both the $k$-*known* setting, in which the number $k$ of clusters is known to learner, and the $k$-*unknown* setting, in which it is not. It is easy to see that the query complexity of every problem in the $k$-unknown setting is no smaller than that of the same problem in the $k$-known setting, since any learning algorithm could simply ignore the value of $k$. We design separate algorithms for the $k$-known and the $k$-unknown setting, while the lower bounds, which we only prove for the $k$-known setting, apply directly to the $k$-unknown setting.

In the next section we describe a number of algorithmic results to solve variants of the $\ell$-PL problem. In particular, Theorem 1 describes the performance of an algorithm for the weighted $\ell$-PL problem in the $k$-known setting. We briefly comment that it is actually not necessary for the number of false negatives $\ell_{\text{no}}$ to be bounded or known in advance in order to run this algorithm and achieve optimal query complexity. Rather the value of $\ell_{\text{no}}$ that appears in the query complexity analysis will be the true number of false-negative responses that occur during the execution of the algorithm. In particular, this implies that in the version of the problem in which no false positives can occur, one does not need to set any upper bound on the error at all. In contrast in the $k$-unknown setting, in order to achieve the bound described in Theorem 3 our proposed algorithm does require a user-chosen value for $\ell_{\text{no}}$. Here, a simple argument shows that it is impossible to guarantee one has found the correct answer unless they can upper bound both $\ell_{\text{yes}}$ and $\ell_{\text{no}}$.

## 4 Algorithmic results

Our first result is an algorithm for the weighted $\ell$-PL problem in the $k$-known setting.

**Theorem 1.** *There exists an algorithm for the weighted $\ell$-PL problem which recovers the full partition $\mathcal{C}$ in the $k$-known setting with query complexity bounded by*

$$O(RS^k(n, k) + (n - k)\ell_{\textit{yes}} + k^2 \ell_{\textit{no}}).$$

We observe an asymmetry in the dependence on $\ell_{\text{yes}}$ and $\ell_{\text{no}}$ respectively, indicating that false-negative and false-positive errors cause the algorithm to incur different costs. In Section 5 we prove lower bounds showing that this is not a mere artifact of the algorithm or its analysis, but rather a fundamental aspect of the problem.

We describe the algorithm and prove Theorem 1 in Section 6. Throughout the paper, when considering the $k$-known setting, we shall assume that $k \notin \{1, n\}$, for otherwise the problem is trivially solved without the need to query the oracle. Because the weighted $\ell$-PL problem generalizes the $\ell$-PL problem, this yields an algorithm for the $\ell$-PL problem as well:

**Corollary 2.** *There exists an algorithm for the $\ell$-PL problem which learns the correct partition in the $k$-known setting with query complexity bounded by*

$$O(RS^k(n,k) + (n-k)\ell + k^2\ell).$$

As our final algorithmic result, in Section D we show that the algorithm can be adapted to obtain optimal asymptotic performance for the $\ell$-PL problem in the $k$-unknown setting.

**Theorem 3.** *There exists an algorithm for the weighted $\ell$-PL problem which recovers the full partition $\mathcal{C}$ in the $k$-unknown setting with query complexity bounded by*

$$O\left(RS^u(n,k) + (n-k)\max\{\ell_{yes}, \ell_{no}\} + k^2\ell_{no}\right).$$

An immediate corollary bounds the query complexity of the $\ell$-PL problem in the $k$-unknown setting.

**Corollary 4.** *There exists an algorithm for the $\ell$-PL problem which learns the correct partition in the $k$-unknown setting with query complexity bounded by*

$$O\left(RS^u(n,k) + \ell(n-k) + k^2\ell\right).$$

In the following section we show that all the bounds discussed in this section are asymptotically optimal. While the results are stated only asymptotically, we note that the upper bounds we obtain are only a small constant factor away from the lower bounds, and we give the exact upper bounds in Section D, where we prove Theorem 3.

## 5 Lower bound techniques: correlation clustering and the chip-liar game

Underlying our lower bound analysis is a connection between partition learning problems, Rényi-Ulam and Chip-Liar games, and correlation clustering. The Rényi-Ulam game, as defined in the background is equivalent to the following Chip-Liar game (see e.g. Chapter 15, in the book of Alon and Spencer [5]). In the game, $N$ chips, numbered 1 to $N$ are placed on a game board that has $\ell + 1$ positions, labeled $0, \ldots, \ell$. At the start of the game, all of the chips begin at position 0 on the board. The game takes place in rounds. On each round the questioner player $Q$ selects a subset $S$ of the chips, and the responder player $R$ decides on one of the following moves: they can either increase the position of every chip in $S$ by 1, or increase the position of every chip in $\overline{S}$ by 1. If a chip at position $\ell$ is in a group whose position is advanced, the chip is said to *have fallen off the board*. After this point such chips will no longer advance. The rules of the game constrain $R$'s responses; $R$ must ensure that on every round, at least one chip remains on the board at position $i \leq \ell$. The game terminates when there is a unique chip remaining on the board.

Note that one can think of the $\ell$-PL problem as a constrained version of this Chip-Liar game. In the $k$-known setting[2], the chips are equivalent to $k$-partitions of the finite set $V$, so that the number $N$ of chips is the Stirling Number of the second kind $N = \left\{ {n \atop k} \right\}$. Moreover, unlike the general Chip-Liar game, in our setup the questioner may only select specific subsets $S$: for any pair of elements $u, v \in V$, the questioner can select $S$ to be the set of all partitions in which $u$ and $v$ are part of the same cluster. When the questioner submits a query ("Are $u$ and $v$ in the same group?"), all chips whose partitions are inconsistent with the response advance by one position on the board.

This allows one to adopt the following perspective. One may think of the queries $\{u_t v_t\}_{t \in [T]}$ made by the questioner together with the signs given by the responses to the queries as making up an instance of the correlation clustering problem. The position of a chip on the board will then be equal the cost of the corresponding partition as a solution of the correlation clustering instance constructed. In Appendix E.1 and Appendix E.2, we formalize this intuition by defining *Rényi-Ulam Correlation Clustering* (RUCC) games, which are the key tools we use to lower bound query complexity.

Leveraging the above techniques, we lower bound the query complexity of the $\ell$-PL$^{\text{FN}}$ and $\ell$-PL$^{\text{FP}}$ problems. Combining these lower bounds with results in the error-free regime yield the following lower bounds on the $\ell$-PL problem.

---

[2]Note that an analogous game can be defined for the $k$-unknown setting, where chips are equivalent to any partition of $V$ and $N$ is the Bell number $B_n$.

**Theorem 5.** *Every algorithm for the $\ell$-PL problem requires at least:*

$$\Omega(\max\{RS(n,k), \ell(n-k), \ell k^2\})$$

*queries both in the $k$-known and in the $k$-unknown setting. Moreover every algorithm for the weighted $\ell$-PL problem requires at least:*

$$\Omega(\max\{RS(n,k), \ell_{yes}(n-k), \ell_{no} k^2\})$$

*queries both in the $k$-known and in the $k$-unknown setting. Here $RS(n,k)$ represents $RS^k(n,k)$ in the $k$-known setting, and $RS^u(n,k)$ in the $k$-unknown setting.*

This theorem is a corollary of Theorems 19 and 21, which we prove in Appendix F.1 and Appendix F.2.

Intuitively, the difference in complexity we observe between the case of false negative errors and the case of false positive errors is due to the following. On one hand, certifying the existence of $k$-clusters using positive answers requires building a spanning forest, which has $(n-k)$ edges. On the other hand, accomplishing the same task using negative answers requires constructing a clique on $k$ vertices which requires $O(k^2)$ edges. This intuition is made precise in Section E in the proofs of Theorem 19 and 21 respectively.

## 6 Algorithm for weighted $\ell$-PL with $k$-known

In this section we give an algorithm that solves the weighted $\ell$-PL problem in the $k$-known setting with asymptotically optimal query complexity. We later adapt this algorithm to the $k$-unknown setting in Appendix D.

We begin by briefly providing intuition for the algorithm. The algorithm maintains $\mathcal{C}'$ a refinement of the correct hidden partition $\mathcal{C}$. At the start of the algorithm $\mathcal{C}'$ is the set of all singleton elements in $V$. The algorithm repeatedly makes sequences of queries that guarantee progress towards either (a) learning $\mathcal{C}$, in the form of establishing that a pair of vertices $u$ and $v$ must be part of the same cluster, or (b) certifying that an error has occurred in the oracle's responses.

In particular, the algorithm repeatedly attempts to construct $(k+1)$-cliques of $-1$ responses. If at any point the oracle responses form such a clique, then a false-negative response must have occurred, as vertices in $V$ belong to at most $k$ distinct clusters (note that $k$ is assumed to be known to the algorithm). In this case, the algorithm has certified a false-negative response. On the other hand, the only way that this process may be interrupted is if the algorithm receives a $(+1)$ response to some query $uv$. However, if such a positive response is received, repeatedly querying the pair $uv$ will either quickly determine that $u$ and $v$ must be part of the same cluster or it will reveal that an error has occurred. In the first case, the algorithm makes progress towards learning $\mathcal{C}$, and in the second case it has certified the occurrence of an error.

We provide the pseudocode for the algorithm and describe in prose the main functions of each subroutine. The algorithm is comprised of subroutines `Learn`, `Get_New`, `Insert` and `Compare`:

**Learn** This is the core component of the algorithm. It maintains a partition $\mathcal{C}'$ for $V$. Throughout the algorithm, $\mathcal{C}'$ is guaranteed to be a refinement of the true hidden partition $\mathcal{C}$. $\mathcal{C}'$ is initialized as the set of singletons for every element in $V$. The algorithm then repeatedly tries to construct a clique of negative answers, supported on some set $S \subseteq V$ until $|S| = k+1$. It does so by inserting new vertices $v$ into $S$ by calling the subroutine `Insert`. `Insert` may then either make progress towards constructing the clique (by increasing the size of $S$) or towards learning the hidden partition by decreasing the size of $\mathcal{C}'$ (merging two clusters together).

**Get_New** Given $S \subseteq V$ and $\mathcal{C}'$ a partition of $V$, `Get_New` returns an element $v \in V \setminus S$ representing a cluster in $\mathcal{C}'$ that's not yet represented in $S$, i.e. $v \notin [u]_{\mathcal{C}'}$ for any $u \in S$.

**Insert** Given an element $v$ and a subset $S \subseteq V$, `Insert` compares $v$ against every element in $S$ by passing $u$ and $v$ to the function `Compare`. `Compare` returns result $\in \{\pm 1\}$. If `Compare` returns a negative result for each $u \in S$, then `Insert` adds $v$ to the subset $S$. On the other hand, if `Compare` returns a positive result for $v$ and some element $u \in S$, then `Insert` updates the partition $\mathcal{C}'$ to reflect that $u$ and $v$ must be in the same cluster. The latter is done by merging $[u]_{\mathcal{C}'}$ with $[v]_{\mathcal{C}'}$.

**Compare**   Given two vertices $u$ and $v$, `Compare` repeatedly submits query $uv$ to the oracle. `Compare` maintains `count` to record information about the number of positive and negative responses received for this query. If at any point `Compare` has received strictly more negative responses than positive responses, it returns `result` $= -1$.

---

**Algorithm 1:** `Learn`$(V, \alpha, k, \ell_{\texttt{yes}})$

---

**Input:** A finite set $V$, an $(\ell_{\texttt{yes}}, \ell_{\texttt{no}})$-faulty oracle $\alpha_{\mathcal{C}}$, a target number of clusters $k$, and error parameters $\ell_{\texttt{yes}}$.
**Output:** A partition $\mathcal{C}'$ of $V$.

1   $S \leftarrow \{\}, \mathcal{C}' \leftarrow \{\{v\} \mid v \in V\}$
2   **while** $|\mathcal{C}'| > k$ **do**
3      $v \leftarrow$ `Get_New`$(S, \mathcal{C}')$
4      $\mathcal{C}', S \leftarrow$ `Insert`$(V, \alpha, S, \mathcal{C}', \ell_{\texttt{yes}})$
5      **if** $|S| = k + 1$ **then**
6         $S \leftarrow \{\}$
7      **end**
8   **end**
9   **return** $\mathcal{C}'$

---

**Algorithm 2:** `Get_New`$(S, \mathcal{C}')$

---

**Input:** A partition $\mathcal{C}'$ of some finite set $V$, and a subset $S \subset V$ such that $|S| \leq |\mathcal{C}'|$
**Output:** An element $v \in V$ representing a set in $\mathcal{C}'$ which is currently not represented by any element of $S$.

1   **for** $v \in V$ **do**
2      **if** $[v]_{\mathcal{C}'} \cap S = \emptyset$ **then**
3         **return** $v$
4      **end**
5   **end**
6   **return** $\perp$

---

**Algorithm 3:** `Insert`$(v, \alpha, \mathcal{C}', S, \ell_{\texttt{yes}})$

---

**Input:** A finite set $V$, a same-cluster oracle $\alpha$ for $V$, a partial clique $S$, a candidate partition $\mathcal{C}'$ of $V$, and error parameter $\ell_{\texttt{yes}}$.
**Output:** A new candidate cluster $\mathcal{C}'$, partial clique $S$.

1   **for** $u \in S$ **do**
2      // Returns either `result` $= -1$ or $+1$
3      `result` $\leftarrow$ `Compare`$(u, v, \alpha, \ell_{\texttt{yes}})$
4      // If $+1$: it is guaranteed that $u$ and $v$ are part of the same cluster and $\mathcal{C}'$ is edited to reflect this
5      **if** $result = +1$ **then**
6         `new_set` $\leftarrow [v]_{\mathcal{C}'} \cup [u]_{\mathcal{C}'}$
7         $\mathcal{C}' \leftarrow (\mathcal{C}' \setminus \{[v]'_{\mathcal{C}}, [u]_{\mathcal{C}'}\}) \cup$ `new_set`
8         **return** $\mathcal{C}', S$
9      **end**
10   **end**
11   // If $S$ is empty or if `result` $= -1$ for every $u \in S$ then a new item is inserted into $S$
12   $S \leftarrow S \cup \{v\}$
13   **return** $\mathcal{C}', S$

---

We briefly remark that for the $k$-unknown setting, Algorithm 9 proposed in the appendix, works in an analogous fashion. The key difference between the two settings is that rather than building cliques of size $k + 1$, Algorithm 9 builds the largest clique possible. We then give a charging argument that shows that this strategy does not lead to a significantly higher query complexity. The algorithm and analysis are presented in Section D.

---
**Algorithm 4:** Compare$(u, v, \alpha, \ell_{\text{yes}})$
---
**Input:** A pair of vertices $u$ and $v$ from some finite set $V$, a same-cluster oracle $\alpha$ for $V$, and error
   parameter $\ell_{\text{yes}}$.
**Output:** `result` $\in \{\pm 1\}$.
---
1  // Initialize `count` to track "how many more times have we observed $r = +1$ compared to
   $r = -1$"
2  `count` $\leftarrow 0$
3  **while** `count` $\geq 0$ **do**
4  |   $r \leftarrow \alpha(u, v)$
5  |   **if** $r = -1$ **and** `count`$> 0$ **then**
6  |   |   `count` $\leftarrow$`count`$-1$
7  |   **else if** $r = -1$ **and** `count`$= 0$ **then**
8  |   |   /* If the number of $-1$ responses exceeds the number of $+1$ responses seen so far, we
   |   |      return $-1$*/
9  |   |   **return** $-1$
10 |   **else**
11 |   |   `count` $\leftarrow$`count`$+1$
12 |   **end**
13 |   // If `count`$/\ell_{\text{yes}} > 1$ then we have verified that $u \sim_{\mathcal{C}*} v$ so return `result` $= +1$
14 |   **if** `count`$/\ell_{yes} > 1$ **then**
15 |   |   **return** $1$
16 |   **end**
17 **end**
---

## 7   Conclusions, Limitations, and Future Directions

In this paper we initiated the study of learning partitions from same-cluster queries subject to bounded adversarial errors. We completely characterize the query complexity of the core problem as well as some of its relevant variants. To do so, we introduce a novel Rényi-Ulam style game based on correlation clustering.

The aim of this paper is to resolve the query complexity of exact recovery with a fixed error tolerance. In particular, the results in this paper do not apply to any setting in which the error may exceed the tolerance $\ell$, e.g. when errors occur probabilistically and independently on every query. This limitation is analogous to those encountered when considering the Hamming (worst-case, bounded) model of error in the theory of error-correcting codes (see e.g. [24]). Thus, like the theory for bounded-error codes, results in this paper should be assumed to hold verbatim in stochastic error settings.

The results in this paper open up several avenues for future research. First, while we settle the complexity of all the variants of the problem in the $k$-known setting, the exact complexity of weighted $\ell$-PL problem in the setting of $k$-unknown remains to be determined. Specifically, the lower bounds of $\Omega(k^2\ell_{\text{no}})$ and $\Omega((n - k)\ell_{\text{yes}})$ that we prove for $k$-known directly extend to the $k$-unknown setting. However, the analysis of Algorithm 9 can be used to establish upper bounds with terms $O(k^2\ell_{\text{no}} + (n - k)\max\{\ell_{\text{yes}}, \ell_{\text{no}}\})$. The question of whether the upper bounds can be tightened to $O(k^2\ell_{\text{no}} + (n - k)\ell_{\text{yes}})$, or whether the corresponding lower bound can be strengthened to e.g. $\Omega((n - k)\max\{\ell_{\text{yes}}, \ell_{\text{no}}\})$ in the $k$-unknown setting, remains open. Another interesting open question is whether there exist algorithms that can achieve exact recovery in the setting of $k$-known when the number of false-positives, $\ell_{\text{yes}}$, is *not* known to the algorithm a priori. In this work we established that such algorithms exist without knowing $\ell_{\text{no}}$ in advance, but it is unclear whether $\ell_{\text{yes}}$ has analogous properties.

We focus on same-cluster queries due to their popularity and simplicity, but there are other practical query models that may interest the community. One such example includes the triangle queries introduced by Vinayak and Hassibi [37] in which a worker is asked to input all pairwise same-cluster responses for three data points at a time. It would be interesting to explore whether the analysis techniques introduced in this work can determine the query complexity of learning partitions under other query models.

**Acknowledgements.** The authors wish to thank Daniel Di Benedetto for providing helpful advice early on in the project; Yibo Jiang for suggesting related work; Mark Olson, Angela Wang, and Nathan Waniorek for useful discussions. AD is supported by NSF DGE 2140001.

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

# A   Additional relevant background

**Comparison of random and adversarial error**   Most of the prior work on clustering with faulty same-cluster queries focuses on the random-error model. Different models of random error have been considered. The most similar works to this paper are the papers of Gupta et al. [23] and Chen et al. [15]. These papers consider non-persistent random error. In this setting, the best known upper bounds on the query complexity of partition learning are

$$\tilde{O}\left(\frac{nk}{f(p)}\right)$$

where $p$ is the probability of error and $f$ is some function that tends to 0 as $p \to 1/2$ [15, 23]. It is difficult to compare the scaling of query complexity under random error as a function of probability of error versus the scaling of query complexity under adversarial error as a function of parameter $\ell$. However one interesting qualitative distinction is that in this work the tight upper and lower bounds on query complexity under adversarial error never pay for a term as large as $n \cdot k \cdot \ell$. Instead in the setting of adversarial error, query complexity bounds scale with $(n + k^2) \cdot \ell$.

**Comparison of persistent and non-persistent error**   Prior literature has also studied the persistent random error version of this problem. The seminal work of Mazumdar and Saha [30] inspired a series of followup works in this setting [16, 22]. For this problem, upper and lower bounds which match up to log factors are known: Mazumdar and Saha [30] establish a lower bound stating that any algorithm achieving probability of recovery at least 3/4 must make at least

$$\Omega\left(\frac{nk}{(1-2p)^2}\right)$$

queries. They further give a (time-inefficient) algorithm which recovers the maximum-likelihood clustering estimate with high probability when $p < 1/2$, using

$$O\left(\frac{nk\log(n)}{(1-2p)^2}\right)$$

queries [30]. In particular, while a piori the persistent-error versions of the problem can be no easier than the non-persistent-error ones, we are not aware of any result that shows a gap in query complexity between these two different settings.

We note that a model imposing persistent adversarial error does not allow for any non-trivial algorithmic results.

**Active learning of graphs with oracles**   The field of graph learning focuses on learning graph structure with oracle access to the graph in question.[3] Numerous settings have been considered in the literature, including instances where the target graphs are matchings [6], stars or cliques [4], or Hamiltonian cycles [35]. Further research examines Las Vegas algorithms [1] and the role that adaptivity plays in the query complexity of these problems [14, 21]. In a related paper, Reyzin and Srivastava [33] give an algorithm that learns the partitions of a graph with $n$ vertices using $O(nk)$ shortest path queries, as well as an algorithm that learns such partitions using $O(n \log n)$ edge counting queries. Angluin and Chen [7] give an algorithm that can learn a graph with $n$ vertices using $O(\log n)$ edge detection queries per edge. Following this line of inquiry, Liu and Mukherjee [26] find that the exact worst-case number of queries needed to learn a partition of $[n]$ into $k$ clusters in the same-cluster query model is exactly $n(k-1) - \binom{k}{2}$ if $k$ is known by the learner a priori, and $nk - \binom{k+1}{2}$ otherwise.

**Geometric clustering with oracle advice**   Ashtiani et al. [8] introduce the problem of clustering data when the learner can make limited same-cluster queries, and they show that this leads to a polynomial-time algorithm for k-means clustering under certain geometric assumptions. In recent

---

[3]In this paper, we invoke the phrase "graph learning" in reference to problems which seek to exactly recover graph structure by actively making queries to some graph oracle, as opposed to the wide field of problems pertaining to learning graphs from data, which often focus on approximate or probabilistic results (i.e. such topics as surveyed by [38]).

years, the model of Ashtiani *et al.* has inspired a variety of other related works which aim to understand the benefits of accessing a same-cluster oracle (e.g. [12, 13, 16]). Ailon et al. [3] show that making a small number of queries to a same-cluster oracle allows one to improve exponentially the dependency on the approximation parameter $\varepsilon$ and on the number $k$ of clusters for polynomial time approximation schemes for some versions of correlation clustering. Saha and Subramanian [34] also consider the problem of using same-cluster queries to aid the solution of a correlation clustering instance, and they give algorithms with performance guarantees that are functions of the optimal value of the input instance.

## B  Equivalence between weighted problem and doubly constrained one

In this section, we argue that, up to changing the parameters by a factor of at most 2, the weighted $\ell$-PL problem is equivalent to one in which the oracle may return up to $\ell_{\text{yes}}$ many false positive answers and $\ell_{\text{no}}$ many false negative ones. We refer to this problem as the asymmetric $\ell$-PL problem.

**Definition 4** (Asymmetric $\ell$-PL Problem). Given a finite set $V$, and a partition $\mathcal{C}$ of $V$, an $(\ell_{\text{yes}}, \ell_{\text{no}})$-asymmetric oracle for $\mathcal{C}$ is an algorithm $\alpha_{\mathcal{C}}$ which, given as input a pair of elements $uv$, returns a value $r = \in \{\pm 1\}$ so that for any sequence of queries $\{(u_t, v_t)\}_{t \in [T]}$ the sequence of responses $\{r_t = \alpha_{\mathcal{C}}(u_t v_t)\}_{t \in [T]}$ satisfies::

$$|\{t \mid r_t = 1 \wedge u_t \not\sim_{\mathcal{C}} v_t\}| \leq \ell_{\text{yes}} \qquad \text{and} \qquad |\{t \mid r_t = -1 \wedge u_t \sim_{\mathcal{C}} v_t\}| \leq \ell_{\text{no}}.$$

The asymmetric $\ell$-PL problem is the problem of recovering $\mathcal{C}$ by making queries to an $(\ell_{\text{yes}}, \ell_{\text{no}})$-asymmetric oracle.

It is easy to see that any $(\ell_{\text{yes}}, \ell_{\text{no}})$-faulty oracle is also an $(\ell_{\text{yes}}, \ell_{\text{no}})$-asymmetric oracle and hence the query complexity of the asymmetric $\ell$-PL problem is no larger than that of the weighted $\ell$-PL problem for the same choice of parameters. Conversely, we have the following:

**Proposition 6.** *For any algorithm which solves the weighted $\ell$-PL problem by making at most $q(n, k, \ell_{yes}, \ell_{no})$ queries, there exists an algorithm which solves the asymmetric $\ell$-PL problem by making at most $q(n, k, 2\ell_{yes}, 2\ell_{no})$ queries.*

*Proof.* This simply follows from the fact that any $(\ell_{\text{yes}}, \ell_{\text{no}})$-asymmetric oracle is also an $(\ell_{\text{yes}}/2, \ell_{\text{no}}/2)$-faulty oracle. $\qquad\square$

## C  Deferred proofs from Section 6

**Query Accounting**   For the sake of analysis, we will assume that the algorithm maintains a global history of all the queries made to the oracle and the responses received. We will also let the algorithm assign labels to query-response pairs. This is done in order to charge the queries made to some measure of progress towards either learning the partitions or certifying that errors have occurred. The lines of pseudocode assigning labels to the query history, written in **blue**, are not necessary for the algorithm to work correctly but rather are included only to facilitate the analysis.

**Lemma 7.** *Throughout the execution of Algorithm 5, $\mathcal{C}'$ is always a refinement of the true partition.*

**Proof of Lemma 7.** $\mathcal{C}'$ is essentially a global variable. We begin by noting that $\mathcal{C}'$ is trivially a refinement of the correct partition at the beginning of the algorithm. The only point in the algorithm at which $\mathcal{C}'$ is modified is on Lines 7 and 8 of `Insert`. In that step, the algorithm modifies $\mathcal{C}'$ by merging the cluster containing $v$ with the cluster containing $u$. This lines are executed only if the value of the variable `result`, output by `Compare`, is $+1$. This happens only if $\text{count}/\ell_{\text{yes}} > 1$ on Line 16 of `Compare`, in which case the latest call to `Compare` has obtained more than $\ell_{\text{yes}}$ many $uv$ queries returning $+1$.

We now argue that if that happens, $u$ and $v$ must have been part of the same cluster in the ground-truth partition $\mathcal{C}$. If $u$ and $v$ were not part of the same of the same cluster in $\mathcal{C}$, we derive a contradiction, as this would imply:

$$\text{FP} \geq \text{count} > \ell_{\text{yes}}$$

and hence $\text{FP}/\ell_{\text{yes}} > 1$. Hence, if $\mathcal{C}'$ was a refinement of $\mathcal{C}$ before merging the clusters containing $u$ and $v$, then it is still a refinement of $\mathcal{C}$ after merging the two clusters. This invariant is then maintained throughout the algorithm, and the lemma holds. $\qquad\square$

---

**Algorithm 5:** $\texttt{Learn}(V, \alpha, k, \ell_{\texttt{yes}})$

---

**Input:** A finite set $V$, an $(\ell_{\texttt{yes}}, \ell_{\texttt{no}})$-faulty oracle $\alpha_{\mathcal{C}}$, a target number of clusters $k$, and error parameters $\ell_{\texttt{yes}}$.

**Output:** A partition $\mathcal{C}'$ of $V$.

**1** $S \leftarrow \{\}, \mathcal{C}' \leftarrow \{\{v\} \mid v \in V\}$
**2** **while** $|\mathcal{C}'| > k$ **do**
**3** $\quad$ $v \leftarrow \texttt{Get\_New}(S, \mathcal{C}')$
**4** $\quad$ $\mathcal{C}', S \leftarrow \texttt{Insert}(V, \alpha, S, \mathcal{C}', \ell_{\texttt{yes}}, \ell_{\texttt{no}})$
**5** $\quad$ **if** $|S| = k + 1$ **then**
**6** $\quad\quad$ $S \leftarrow \{\}$
**7** $\quad$ **end**
**8** **end**
**9** **return** $\mathcal{C}'$

---

---

**Algorithm 6:** $\texttt{Get\_New}(S, \mathcal{C}')$

---

**Input:** A partition $\mathcal{C}'$ of some finite set $V$, and a subset $S \subset V$ such that $|S| \leq |\mathcal{C}'|$

**Output:** An element $v \in V$ representing a set in $\mathcal{C}'$ which is currently not represented by any element of $S$.

**1** **for** $v \in V$ **do**
**2** $\quad$ **if** $[v]_{\mathcal{C}'} \cap S = \emptyset$ **then**
**3** $\quad\quad$ **return** $v$
**4** $\quad$ **end**
**5** **end**
**6** **return** $\perp$

---

---

**Algorithm 7:** $\texttt{Insert}(v, \alpha, \mathcal{C}', S, \ell_{\texttt{yes}})$

---

**Input:** A finite set $V$, a same-cluster oracle $\alpha$ for $V$, a partial clique $S$, a candidate partition $\mathcal{C}'$ of $V$, and error parameters $\ell_{\texttt{yes}}$.

**Output:** A new candidate cluster $\mathcal{C}'$, partial clique $S$.

**1** **for** $u \in S$ **do**
**2** $\quad$ // Returns either $\texttt{result} = -1$ or $+1$
**3** $\quad$ $\texttt{result} \leftarrow \texttt{Compare}(u, v, \alpha, \ell_{\texttt{yes}}, \ell_{\texttt{no}})$
**4** $\quad$ // If $+1$: it is guaranteed that $u$ and $v$ are part of the same cluster and $\mathcal{C}'$ is edited to reflect this
**5** $\quad$ **if** $\texttt{result} = +1$ **then**
**6** $\quad\quad$ Re-label all 'clique' queries made during this execution of $\texttt{Insert}$ (and all calls to $\texttt{Compare}$ therein) as 'merge-'
**7** $\quad\quad$ $\texttt{new\_set} \leftarrow [v]_{\mathcal{C}'} \cup [u]_{\mathcal{C}'}$
**8** $\quad\quad$ $\mathcal{C}' \leftarrow (\mathcal{C}' \setminus \{[v]'_{\mathcal{C}}, [u]_{\mathcal{C}'}\}) \cup \texttt{new\_set}$
**9** $\quad\quad$ **return** $\mathcal{C}', S$
**10** $\quad$ **end**
**11** **end**
**12** // If $S$ is empty or if $\texttt{result} = -1$ for every $u \in S$ then a new item is inserted into $S$
**13** $S \leftarrow S \cup \{v\}$
**14** **return** $\mathcal{C}', S$

---

---

**Algorithm 8:** Compare$(u, v, \alpha, \ell_{\texttt{yes}})$

---

**Input:** A pair of vertices $u$ and $v$ from some finite set $V$, a same-cluster oracle $\alpha$ for $V$, and error parameters $\ell_{\texttt{yes}}$.

**Output:** result $\in \{\pm 1\}$.

1   // Initialize count to track "how many more times have we observed $r = +1$ compared to $r = -1$"

2   count $\leftarrow 0$

3   **while** *count* $\geq 0$ **do**

4     $r \leftarrow \alpha(u, v)$

5     **if** $r = -1$ ***and*** *count* $> 0$ **then**

6       count $\leftarrow$ count$-1$

7     **else if** $r = -1$ ***and*** *count* $= 0$ **then**

8       /* If the number of $-1$ responses exceeds the number of $+1$ responses seen so far, we return $-1$*/

9       Label the last query as 'clique'

10      Label all other queries created on this execution of Compare as 'spurious'

11      **return** $-1$

12     **else**

13       count $\leftarrow$ count$+1$

14     **end**

15     // If count$/\ell_{\texttt{yes}} > 1$ then we have verified that $u \sim_{\mathcal{C}*} v$ so return result $= +1$

16     **if** *count*$/\ell_{yes} > 1$ **then**

17       Label the last count-many queries which received positive responses as 'merge+'

18      Label all other queries made during this execution of Compare as 'spurious'

19      **return** 1

20     **end**

21 **end**

---

**Lemma 8.** *At the end of the algorithm, the history of queries contains at most $(\ell_{yes} + 1)(n - k)$ queries labelled 'merge+'and at most $k(n - k)$ queries labelled 'merge-'.*

**Proof of Lemma 8.** Consider a point in the algorithm in which Compare returns $+1$. When that happens, the **if** block starting on Line 5 of Insert is executed; we refer to this as a *merge event*.

We begin by arguing that at most $n - k$ merge events can occur during an execution of Algorithm 5. At the start of the algorithm $|\mathcal{C}'| = n$. At each merge event two clusters in $\mathcal{C}'$ are merged and the cardinality of $\mathcal{C}'$ decreases by 1. Since $\mathcal{C}'$ is always a refinement of the hidden $k$-partition $\mathcal{C}$ (by Lemma 7), this implies that at most $n - k$ merge events can occur during the execution of the algorithm.

Both 'merge-' and 'merge+' queries are only created in correspondence with a merge event. We now show that on any merge event, at most $k$ 'merge-' queries and at most $\ell_{\texttt{yes}} + 1$ 'merge+' queries are created. By the above argument these results imply the statement of the lemma.

We now show that at most $k$ 'merge-' queries are created during each merge event. 'merge-' queries are created when some 'clique' queries are re-labelled as 'merge-'. Specifically, all the 'clique' queries created during every execution of Compare within the current execution of Insert are relabelled as 'merge-'. On any execution of Insert, at most one 'clique' query is created for every element in the current set $S$. During any iteration of the **for** loop at Insert Line 1, the set $S$ is of size at most $k$, implying that at most $k$ 'clique' queries are created during one execution of Insert. As a result at most $k$ 'merge-' queries are created at every merge event.

Similarly, 'merge+' queries are only created in correspondence with a merge event. Specifically, merge events occur when Compare returns result$= +1$. When this happens, 'merge+' queries are created at Compare Line 17. When Line 17 is executed, the number of queries labelled 'merge+' is equal to the current value of count, which is at most at most $\ell_{\texttt{yes}} + 1$. Therefore, at most $\ell_{\texttt{yes}} + 1$ 'merge+'queries are created in correspondence with each merge event, concluding the proof.   □

**Lemma 9.** *At the end of the algorithm, the history of queries contains at most* $2 \max\{\ell_{yes}, \ell_{no}\}$ *queries labelled* '`spurious`'.

**Proof of Lemma 9.** Each query $uv$ labelled '`spurious`' can be paired uniquely with another query $uv$ labelled `spurious` which returned the opposite answer. Hence, the number of erroneous answers returned by the oracle is at least the number of `spurious` queries divided by two. This number is upper bounded by $\max\{\ell_{yes}, \ell_{no}\}$ and the result follows. $\square$

**Lemma 10.** *At the end of the algorithm, the history of queries contains at most* $(\ell_{no} + 1)\binom{k+1}{2}$ *queries labelled* '`clique`'.

**Proof of Lemma 10.** The algorithm only labels queries as '`clique`' in Line 9 of `Compare`. Thus every '`clique`' query is made between two elements $u$ and $v$ such that both $u$ and $v$ were inserted into some construction of a clique, supported on the set $S$.

If $S$ reaches cardinality $k + 1$, the **if** block on Line 5 of `Insert` is executed. We refer to this occurrence as a *clique event*. When a clique event occurs, queries have made up new a clique of $-1$ responses of size $k + 1$. In order for all of these responses to have been correct, the ground-truth partition would need to have been size at least $k + 1$, hence every time a clique event happens, a false negative error must have occurred. Since every clique event can be charged to a false negative error, at most $\ell_{no}$ clique events can occur.

We will charge every '`clique`' query made before a clique event to the first clique event following the query's creation. These queries represent a distinct edge $uv$ of a clique built on $S$ which eventually reaches size $k + 1$, and hence exactly $\binom{k+1}{2}$ many of these '`clique`' queries will occur per clique event. Finally, all '`clique`' queries occurring after the last clique event make up the edges of a clique supported on the set $S$, which never reaches size $k + 1$, and hence there are less than $\binom{k+1}{2}$ such queries.

The result then follows. $\square$

**Lemma 11.** *If Algorithm 5 makes a finite number of queries, then it must terminate.*

**Proof of Lemma 11.** We show that if Algorithm 5 makes a finite number of queries, then Line 4 of Algorithm 5 is executed a finite number of times. This implies termination of Algorithm 5.

When Line 4 of Algorithm 5 executes, Algorithm 5 calls Algorithm 7, `Insert`. If $S$ is nonempty on the execution of Line 4, then `Insert` calls `Compare` on some pair of elements. On every call to `Compare`, Line 4 of `Compare` executes at least once, and hence the algorithm makes at least one query. If $S$ is empty on the execution of Line 4, then `Insert` increases the size of $S$ by 1. This implies that on the next execution of Line 4, $S$ will be nonempty. Thus in order for Algorithm 5 to execute Line 4 infinitely many times, the algorithm must make infinitely many queries, yielding the result. $\square$

We now analyze the number of queries made by the algorithm. We begin by noting that when the algorithm terminates, every query made will hold exactly one of the following labels: '`merge-`', '`merge+`', '`clique`', '`spurious`'. The analysis proceeds by showing how labels of each category contribute to either (a) correctly identifying the true partition, or (b) correctly certifying the occurrence of errors.

Equipped with these lemmata, we now prove Theorem 1.

*Proof of Theorem 1.* Let $|\text{'merge-'}|, |\text{'merge+'}|, |\text{'spurious'}|, |\text{'clique'}|$ be the number of '`merge-`', '`merge+`', '`spurious`' and '`clique`' queries made by the algorithm respectively. By Lemmata 8, 9, and 10, the algorithm makes at most

$$|\text{'merge-'}| + |\text{'merge+'}| + |\text{'spurious'}| + |\text{'clique'}|$$

$$\leq nk - k^2 + (\ell_{yes} + 1)(n - k) + 2\max\{\ell_{yes}, \ell_{no}\} + (\ell_{no} + 1)\binom{k+1}{2}$$

$$= O\left(RS^k(n, k) + (n - k)\ell_{yes} + k^2 \ell_{no}\right)$$

many queries. By Lemma 11 this implies that the algorithm must terminate.

When the algorithm terminates, the condition in the while loop of Learn is not met, and hence $|\mathcal{C}'| \leq k$. But since, by Lemma 7 $\mathcal{C}'$ is a refinement of $\mathcal{C}$, it must hold that upon termination $|\mathcal{C}'| = k$ and hence $\mathcal{C}' = \mathcal{C}$. The algorithm then returns the correct hidden partition $\mathcal{C}$. $\qquad\square$

# D  The algorithm for $k$-unknown

In this section we describe an algorithm for the weighted $\ell$-PL problem in the $k$-unknown setting, and further prove that it achieves optimal asymptotic performance for the unweighted version of the problem. A complete description of the algorithm is given in the pseudocode below. Note that the algorithm makes use of subroutines defined in Section D. The main challenge in adapting Algorithm 5 in Section 6 is that the algorithm crucially relies on knowing the value of $k$ to build a lower bound to the number of observed errors, by counting the number of $(k + 1)$-cliques of $-1$ answers observed.

To overcome this problem, the new algorithm will instead simply aim to build the largest possible clique of $-1$ answers. We show that, remarkably, when false positive and false negative errors are equivalent ($\ell_{\texttt{yes}} = \ell_{\texttt{no}} = \ell$), this suffices to match the optimal guarantees of Algorithm 5, up to small constant factors.

---

**Algorithm 9:** Learn_k_unknown$(V, \alpha, \ell_{\texttt{yes}}, \ell_{\texttt{no}})$

**Input:** A finite set $V$, a same-cluster oracle $\alpha$, error parameters $\ell_{\texttt{yes}}$ and $\ell_{\texttt{no}}$.
**Output:** A partition $\mathcal{C}'$ of $V$.

1   $\mathcal{C}' \leftarrow \{\{v\} \mid v \in V\}$, idx$\leftarrow 1$
2   $S_{\texttt{idx}} \leftarrow \{\}$
3   **while** $idx \leq \ell_{no} + 1$ **do**
4      $v \leftarrow$ Get_New$(S, \mathcal{C}')$
5      **if** $v == \bot$ **then**
6         idx$+ = 1$
7         $S_{\texttt{idx}} \leftarrow \{\}$
8      **end**
9      **else**
10         $\mathcal{C}', S_{\texttt{idx}} \leftarrow$ Insert$(V, \alpha, S_{\texttt{idx}}, \mathcal{C}', \ell_{\texttt{yes}})$
11      **end**
12   **end**
13   **return** $\mathcal{C}'$

---

The intuition behind the success of Algorithm 9 is that, for the algorithm to construct large cliques (significantly larger than the cardinality $k$ of the hidden partition), the oracle must return many incorrect answers. We formalize this intuition in Lemma 12.

Consider the execution of Algorithm 9. The algorithm will repeatedly construct cliques of $-1$ queries supported on vertex sets $S_i$, adding elements to $S_i$ until all elements of $V$ belong to an equivalence class $[u]_{\mathcal{C}'}$ for some $u \in S_i$. Once this condition is met, the algorithm increments $i$ and initializes $S_{i+1}$ as the empty set. The following lemma shows that the maximum size of any such set $S_i$ constructed by the algorithm is bounded as a function of the true unknown partition size $k$ and the error parameter $\ell_{\texttt{no}}$.

Consider the set $S_i$ constructed by Algorithm 9 while idx $= i$. We will denote by $n_i$ the maximum cardinality attained by this set $S_i$. For convention we will also set $n_0 = n$. Moreover, we let $\ell_i$ denote the number of false negative responses returned while idx $= i$.

**Lemma 12.** *For any $i \geq 1$, we have:*

$$n_i \leq k + \sqrt{2k\ell_i}. \tag{3}$$

*Proof.* Note that, if the hidden partition has cardinality $k$, and the algorithm obtains a clique of $-1$ responses on $n_i$ vertices, then the number of false negative errors contained in the clique responses is at least the minimum number of within-cluster edges of any $k$-partition of the complete graph $K_{n_i}$.

I.e. a lower bound to the number of false negative errors that must have occurred is given by:

$$\ell_i \geq \min\left\{\sum_{j=1}^{k}\binom{c_j}{2}\,\middle|\,c_1,...,c_k \in \mathbb{N}, \sum_{j=1}^{k}c_j = n_i\right\}$$

$$\geq \min\left\{\sum_{j=1}^{k}\frac{c_j(c_j-1)}{2}\,\middle|\,c_1,...,c_k \in \mathbb{R}_{>0}, \sum_{j=1}^{k}c_j = n_i\right\}.$$

Note that the function $f(x) = \frac{x(x-1)}{2}$ is convex, and hence by Karamata's Inequality (Theorem 1 from [25]) the right-hand side above is lower bounded by:

$$\ell_i \geq \sum_{j=1}^{k}\frac{\frac{n_i}{k}\left(\frac{n_i}{k}-1\right)}{2} = \frac{1}{2}n_i\left(\frac{n_i}{k}-1\right).$$

Re-arranging, we find that $n_i^2 - kn_i \leq 2\ell_i k$. Completing the square we obtain:

$$\left(n_i - \frac{1}{2}k\right)^2 \leq 2\ell_i k + \frac{1}{4}k^2$$

giving:

$$n_i - \frac{1}{2}k \leq \sqrt{2\ell_i k + \frac{1}{4}k^2} \leq \sqrt{2\ell_i k} + \frac{1}{2}k,$$

and hence:

$$n_i \leq \sqrt{2\ell_i k} + k,$$

as needed. $\qquad\square$

Crucially, we note that as with Algorithm 5, the partition $\mathcal{C}'$ maintained by Algorithm 9 is always a refinement of the ground-truth.

**Lemma 13.** *Throughout the execution of Algorithm 9, $\mathcal{C}'$ is always a refinement of the true partition.*

The proof of this lemma is identical to the proof of Lemma 7.

**Lemma 14.** *Algorithm 9 terminates.*

*Proof.* We show that Lines 6 and 10 in Algorithm 9 execute finitely many times. This implies termination of the algorithm.

Every time Line 6 executes, the value of idx is incremented by 1. Once the value of idx exceeds $\ell_{\texttt{no}} + 1$, the **while** loop in Algorithm 9 exits, and thus Line 6 can only execute finitely many times.

To show that Line 10 executes finitely many times, we show that every time Line 10 executes, either $\mathcal{C}'$ decreases in cardinality or $S_{\texttt{idx}}$ increases in cardinality. This follows from considering Algorithm 7, Insert. On each call to Insert, either result $= +1$ for some $u \in S$, in which case the cardinality of $\mathcal{C}'$ is reduced in Line 8, or result $= -1$ for every $u \in S$. In the latter case, the size of $S$ is increased in Line 13.

By Lemma 13, $\mathcal{C}'$ is always a refinement of $\mathcal{C}$ and thus its cardinality cannot decrease more than $(n - k)$ times. Similarly, for each value idx $= i$, $|S_i| \leq n$ so $S_i$ can increase in size finitely many times for each value of idx $\in [1, \ell_{\texttt{no}} + 1]$. This implies that Line 10 can only execute finitely many times, completing the proof. $\qquad\square$

**Lemma 15.** *When Algorithm 9 terminates $\mathcal{C}'$ is equal to the ground-truth partition $\mathcal{C}$.*

*Proof.* By Lemma 13 $\mathcal{C}'$ is always a refinement of $\mathcal{C}$. Therefore to show that the partition $\mathcal{C}'$ returned is equal to $\mathcal{C}$, it remains to show that the cardinality of $\mathcal{C}'$ upon termination is the true unknown value $k$.

Assume for the sake of contradiction that this does not hold. Once initialized, the only place where $\mathcal{C}'$ is modified during the execution of Algorithm 9 is in Lines 7 and 8 of Insert. There, the algorithm

modifies $\mathcal{C}'$ by merging two previously disjoint equivalence classes. This reduces the number of disjoint equivalence classes in $\mathcal{C}'$, and so in particular the cardinality of $\mathcal{C}'$ montonically decreases during the execution of Algorithm 9. Thus if upon termination the cardinality of $\mathcal{C}'$ is strictly greater than $k$, then this was true at all points during the algorithm's execution.

Recall that $n_i$ is defined to be the maximum cardinality attained by set $S_i$. In particular, $S_i$ attains maximum size when Line 6 of Algorithm 9 is executed. When this happens, $S_i$ contains exactly one representative from every cluster in $\mathcal{C}'$ at the time of execution. If the size of $\mathcal{C}'$ is strictly greater than $k$ at all points during the execution of Algorithm 9 then $n_i > k \ \forall i \in [0, \ell_{\mathrm{no}} + 1]$.

In particular for every value $i \in [1, \ell_{\mathrm{no}} + 1]$, for every distinct $u, v \in S_i$ Algorithm 9 receives at least one response $-1$ for query $(u, v)$. This response is consistent with the scenario in which $u$ and $v$ belong to disjoint sets in the ground truth partition $\mathcal{C}$. If $n_i > |\mathcal{C}|$ then at least one of these responses must have been a false negative error. As this holds for every value $i \in [1, \ell_{\mathrm{no}} + 1]$, this implies least $\ell_{\mathrm{no}} + 1$ false negative responses occur during the execution of Algorithm 9. In particular this violates the assumption that $\alpha$ is an $(\ell_{\mathrm{yes}}, \ell_{\mathrm{no}})$-faulty oracle, and we thus conclude that the size of $\mathcal{C}'$ cannot be greater than $k$ upon termination, implying that the partition returned by the algorithm satisfies $\mathcal{C}' = \mathcal{C}$. $\qquad \square$

As with the analysis of Algorithm 5, bounding the query complexity of Algorithm 9 reduces to bounding the number of queries with labels 'merge-', 'merge+', 'clique', 'spurious' upon termination of the algorithm.

**Lemma 16.** *When Algorithm 9 terminates the history of queries contains at most $\left(1 + \sqrt{2}\right)(\ell_{no} + 1)k^2$ queries labelled 'clique'.*

*Proof.* All queries labeled 'clique' upon termination must have been created during the construction of some set $S_i$ while idx$= i$. Given $n_i$ the maximum cardinality attained by set $S_i$, the number of 'clique' queries present upon termination that were constructed while idx$= i$ is exactly

$$\binom{n_i}{2} = \frac{1}{2}(n_i^2 - n_i) \leq k^2 + 2k\sqrt{2k\ell_i} + 2k\ell_i - n_i$$

where the upperbound follows from Lemma 12 and $\ell_i$ denotes the number of false negative responses returned by the oracle while idx$= i$. Summing the total number of such queries over all iterates idx$= 1, \ldots, \ell_{\mathrm{no}} + 1$, we obtain:

$$\sum_{i=1}^{\ell_{\mathrm{no}}+1} \binom{n_i}{2} \leq \frac{1}{2} \sum_{i=1}^{\ell_{\mathrm{no}}+1} k^2 + 2k\sqrt{2k\ell_i} + 2k\ell_i - n_i$$

$$= \frac{1}{2}(\ell_{\mathrm{no}}+1)k^2 + \sqrt{2} \sum_{i=1}^{\ell_{\mathrm{no}}+1} k\sqrt{k\ell_i} + k \sum_{i=1}^{\ell_{\mathrm{no}}+1} \ell_i - \frac{1}{2} \sum_{i=1}^{\ell_{\mathrm{no}}+1} n_i$$

$$\leq \frac{1}{2}(\ell_{\mathrm{no}}+1)k^2 + \sqrt{2}k \sum_{i=1}^{\ell_{\mathrm{no}}+1} \frac{\ell_i + k}{2} + k \sum_{i=1}^{\ell_{\mathrm{no}}+1} \ell_i - \frac{1}{2} \sum_{i=1}^{\ell_{\mathrm{no}}+1} n_i$$

$$= \frac{1}{2}(\ell_{\mathrm{no}}+1)k^2 + \frac{k\ell_{\mathrm{no}}}{\sqrt{2}} + \frac{(\ell_{\mathrm{no}}+1)k^2}{\sqrt{2}} + k\ell_{\mathrm{no}} - \frac{1}{2}k(\ell_{\mathrm{no}}+1).$$

The last line follows by observing that by definition of $\ell_i$, $\sum_{i=1}^{\ell+1} \ell_i \leq \ell_{\mathrm{no}}$.

We can simplify the above expression using the fact that for integer $k$ it holds that $k \leq k^2$, to obtain the bound:

$$\frac{1}{2}(\ell_{\mathrm{no}}+1)k^2 + \frac{k\ell_{\mathrm{no}}}{\sqrt{2}} + \frac{(\ell_{\mathrm{no}}+1)k^2}{\sqrt{2}} + k\ell - \frac{1}{2}k(\ell_{\mathrm{no}}+1) = \left(\frac{1}{2} + \frac{1}{\sqrt{2}}\right)\left((\ell_{\mathrm{no}}+1)k^2 + k\ell_{\mathrm{no}}\right)$$

$$\leq \left(1 + \sqrt{2}\right)(\ell_{\mathrm{no}}+1)k^2.$$

$\qquad \square$

**Lemma 17.** *Upon termination of Algorithm 9, the history of queries contains at most $2\max\{\ell_{yes}, \ell_{no}\}$ queries labelled 'spurious'.*

The proof follows analogously to that of Lemma 9.

**Lemma 18.** *Upon termination of Algorithm 9, the history of queries contains at most $(\ell_{yes}+1)(n-k)$ queries labelled 'merge+', and at most*

$$(n-k)\max\left\{(1+\sqrt{2})k, k+\ell_{no}\sqrt{2}\right\}$$

*queries labelled 'merge-'.*

*Proof.* Similarly to the analysis of Algorithm 5, we analyze the numbers of 'merge+' and 'merge-' queries, by considering *merge events*. A merge event is defined as the execution of Line 5 in `Insert`.

Recall that $n_i$ is define to be the maximum cardinality attained by set $S_i$. In particular, $S_i$ attains maximum size when Line 6 of Algorithm 9 is executed. When this happens, $S_i$ contains exactly one representative from every cluster in $\mathcal{C}'$ at the time of execution. Thus $n_i - n_{i+1}$ is equal to the number of clusters merged while $\mathtt{idx} = (i+1)$, i.e. the number of merge events that occurred while $\mathtt{idx} = (i+1)$. Moreover the size of $\mathcal{C}'$ upon termination of Algorithm 9 is equal to $n_{\ell_{no}+1}$, and hence by Lemma 15 $n_{\ell_{no}+1} = k$. This implies

$$\sum_{i=0}^{\ell_{no}} n_i - n_{i+1} = n_0 - n_{\ell_{no}+1} = n - k.$$

As with the proof of Lemma 8, we note that each 'merge+' query can be charged to a unique merge event in such a way that exactly $\ell_{yes} + 1$ 'merge+' queries are charged to each merge event. As a result, the number $|\text{'merge+'}|$ of 'merge+' queries satisfies:

$$|\text{'merge+'}| = \sum_{i=0}^{\ell_{no}} (\ell_{yes} + 1)(n_i - n_{i+1}) = (\ell + 1)(n - k)$$

Similarly, 'merge-' queries are only created in correspondence with a merge event. Specifically, 'merge-' are created when 'clique' queries are re-labelled as 'merge-'. When $\mathtt{idx} = (i+1)$ on any single execution of `Insert`, at most one 'clique' query is created for every element in the current set $S_{(i+1)}$. Hence at most $n_{i+1}$ 'merge-' queries correspond to each merge event that occurs while $\mathtt{idx} = (i+1)$. Summing over all values attained by $\mathtt{idx}$, this implies the total number of queries labeled 'merge-' is bounded by

$$|\text{'merge-'}| \leq \sum_{i=0}^{\ell_{no}} n_{i+1}(n_i - n_{i+1})$$

$$\leq \sum_{i=0}^{\ell_{no}} (k + \sqrt{2k\ell_{i+1}})(n_i - n_{i+1})$$

$$= k\sum_{i=0}^{\ell_{no}} (n_i - n_{i+1}) + \sqrt{2k}\sum_{i=0}^{\ell_{no}} (n_i - n_{i+1})\sqrt{\ell_{i+1}},$$

where the second inequality follows from the bound in Lemma 12. Invoking the fact that $\sum_{i=0}^{\ell_{no}} n_i - n_{i+1} = n - k$, we can bound the above and obtain:

$$|\text{'merge-'}| \leq k \cdot (n - k) + \sqrt{2k} \cdot \max_i \sqrt{\ell_{i+1}} \cdot \sum_{i=0}^{\ell_{no}} (n_i - n_{i+1})$$

$$\leq \left(k + \sqrt{2k\ell_{no}}\right)(n - k)$$

$$\leq (n - k) \cdot \max\left\{(1 + \sqrt{2})k, k + \ell_{no}\sqrt{2}\right\}$$

yielding the result. □

Equipped with the above lemmata, we proceed to prove Theorem 3.

*Proof of Theorem 3.* By Lemmata 14 and 15, Algorithm 9 terminates and returns the correct hidden partition. It thus remains to bound the number of queries made during execution.

Let $|\text{`merge-'}|, |\text{`merge+'}|, |\text{`spurious'}|, |\text{`clique'}|$ be the number of `merge-', `merge+', `spurious' and `clique' queries made by the algorithm respectively. By Lemmata 16, 17, and 18, the algorithm makes at most

$$|\text{`merge-'}| + |\text{`merge+'}| + |\text{`spurious'}| + |\text{`clique'}|$$

$$\leq (n-k)\max\left\{(1+\sqrt{2})k, k + \ell_{\text{no}}\sqrt{2}\right\} + (\ell_{\text{yes}} + 1)(n-k)$$

$$+ 2\max\{\ell_{\text{yes}}, \ell_{\text{no}}\} + (1+\sqrt{2})(\ell_{\text{no}} + 1)k^2$$

$$= O\left(RS^u(n,k) + (n-k)\max\{\ell_{\text{yes}}, \ell_{\text{no}}\} + k^2\ell_{\text{no}}\right)$$

many queries. $\qquad\square$

# E  Lower bound results

## E.1  Lower bounds for the easy problem: $\ell$-false negatives

We prove the following lower bound on the query complexity of the $\ell\text{-PL}^{\text{FN}}$ problem.

**Theorem 19.** *Every algorithm for the $\ell\text{-PL}^{\text{FN}}$ problem which guarantees full recovery of the correct partitions requires $RS^k(n,k) + \Omega(k^2\ell)$ queries in the worst case.*

Because the $\ell\text{-PL}^{\text{FN}}$ problem is no harder than the $\ell\text{-PL}$ problem, this immediately implies the same lower bound on the query complexity of the $\ell\text{-PL}$ problem.

To establish this result, we introduce and analyze a game between a Quetionner and a Responder, representing the algorithm and the oracle respectively.

**Definition of the RUCC$^{\text{FN}}$ game.**  We define the following game, which we call the RUCC$^{\text{FN}}$ game. The game is played by two players, the questioner ($Q$) and the responder ($R$) and it is parametrized by a finite set $V$ of cardinality $n$, a positive integer $k \in \{2, ..., n-1\}$, and a non-negative integer $\ell$. The goal of the questioner is to infer a $k$-partition of $V$ given input from the responder. At the start of the game, we are given a complete undirected graph $G_0$ on vertex set $V$. $G_0 = (V, E, w_0^+, w_0^-)$ has two associated edge weight functions $w_0^+ : E \to \mathbb{R}$ and $w_0^- : E \to \mathbb{R}$, both of which are set to zero at the beginning of the game.

At each iteration $t$, the questioner $Q$ submits a query $uv$ for distinct $u$ and $v$ in $V$. The responder $R$ then issues a response $r_t \in \{\pm 1\}$. If $r_t = +1$, then $w^+$ is updated by setting $w_t^+(uv) = w_{t-1}^+(uv) + 1$. Otherwise, if $r_t = -1$, then $w^-$ is updated by setting $w_t^-(uv) = w_{t-1}^-(uv) + 1$. We then set $G_t = (V, E, w_t^+, w_t^-)$.

For any partition $\mathcal{C}$ of $V$, we define the cost:

$$\text{cost}_{G_t}^{\text{FN}}(\mathcal{C}) \overset{\text{def}}{=} \sum_{\substack{uv \in E \\ u \sim_{\mathcal{C}} v}} w_t^-(uv) + \mathbb{I}\left(\sum_{\substack{uv \in E \\ u \not\sim_{\mathcal{C}} v}} w_t^+(uv) = 0\right),$$

where $\mathbb{I}$ is the convex indicator function. Intuitively, the value of $\text{cost}_{G_t}^{\text{FN}}(\mathcal{C})$ is the number of incorrect negative ($-1$) answers that the responder would have to have given if $\mathcal{C}$ was the correct partition, or infinity if the responder would have had to given any incorrect positive ($+1$) answer.

We require that the responder must, at each iteration $t$, reply in a way that guarantees that there exists some $k$-partition $\mathcal{C}$ of $V$ such that:

$$\text{cost}_{G_t}^{\text{FN}}(\mathcal{C}) \leq \ell.$$

The game terminates when $\text{cost}_{G_t}^{\text{FN}}(\mathcal{C})$ is strictly more than $\ell$ for all but one $k$-partition $\mathcal{C}$ of $V$, indicating that the responder is left with a single feasible partition and hence they have learned the ground truth. If the game terminates at iteration $T$, then the payoff to the responder player is $T$, and the payoff to the questioner is $-T$.

Given a questioner strategy $Q$ and a responder strategy $R$, we let $\text{Game}^{\text{FN}}(Q, R)$ be the number of iterations the game lasts when questioner $Q$ plays against responder $R$, as a function of the parameters $n$, $k$ and $\ell$. The query complexity of the $\ell$-PL$^{\text{FN}}$ problem is the value:

$$\min_{Q \in \mathcal{Q}} \max_{R \in \mathcal{R}} \text{Game}^{\text{FN}}(Q, R),$$

where $\mathcal{Q}$ is the set of all possible questioner strategies and $\mathcal{R}$ is the set of all possible responder strategies. A simple argument then shows:

**Lemma 20.** *Let $f$ be a non-negative real valued function. If there exists a responder strategy $R^*$ such that:*

$$\forall n, k, \ell \in \mathbb{N}_0 : \qquad \min_{Q \in \mathcal{Q}} \text{Game}^{\text{FN}}(Q, R^*) \geq f(n, k, \ell),$$

*then the query complexity of the $\ell$-PL$^{\text{FN}}$ problem is at least $f(n, k, \ell)$.*

*Proof.* This follows from:

$$\min_{Q \in \mathcal{Q}} \max_{R \in \mathcal{R}} \text{Game}^{\text{FN}}(Q, R) \geq \min_{Q \in \mathcal{Q}} \text{Game}^{\text{FN}}(Q, R^*) \geq f(n, k, \ell).$$

$\square$

In Theorem 23 in Appendix F.1 we show that there exists a responder strategy $R^*$ satisfying

$$\min_{Q \in \mathcal{Q}} \text{Game}^{\text{FN}}(Q, R^*) \geq (\ell + 1)\left(\binom{k + 1}{2} - 1\right) = \Omega(\ell k^2)$$

queries. Lemma 20 and Theorem 23, together with the lower bound of Liu and Mukherjee for the problem without errors [26] immediately imply Theorem 19.

### E.2 Lower bounds for the easy problem: $\ell$-false positives

We prove the following lower bound on the query complexity of the $\ell$-PL$^{\text{FP}}$ problem:

**Theorem 21.** *Every algorithm for the $\ell$-PL$^{\text{FP}}$ problem which guarantees exact recovery of the correct partition requires $RS^k(n, k) + \Omega(\ell(n - k))$ queries in the worst case.*

As in Section E.1, this result immediately implies the same lowerbound for the $\ell$-PL problem. In order to analyze the query complexity of learning partitions with $\ell$ false positives, we consider a very similar game to that defined above in Appendix E.1.

**Definition of the RUCC$^{\text{FP}}$ game.** The RUCC$^{\text{FP}}$ game is defined with the same setup as the RUCC$^{\text{FN}}$ game. The questioner Q, responder R, finite set $V$, and gameplay graph $G_t = (V, E, w_t^+, w_t^-)$ are all defined as in Appendix E.1. The core distinction between the RUCC$^{\text{FP}}$ game and the RUCC$^{\text{FN}}$ game is the notion of cost employed. For the RUCC$^{\text{FP}}$ game, for any partition $\mathcal{C}$ of $V$ we define the cost

$$\text{cost}_{G_t}^{\text{FP}}(\mathcal{C}) \stackrel{\text{def}}{=} \sum_{\substack{uv \in E \\ u \not\sim_{\mathcal{C}} v}} w_t^+(uv) + \mathbb{I}\left(\sum_{\substack{uv \in E \\ u \sim_{\mathcal{C}} v}} w_t^-(uv)\right).$$

Intuitively, the value of $\text{cost}^{\text{FP}}(\mathcal{C})$ is either the number of incorrect positive answers $(+1)$ that the responder would have to have given if $\mathcal{C}$ were the correct partition, or the value is infinity if the responder would have given any incorrect negative $(-1)$ answer.

The rules of the game require that at each iteration $t$, the responder must reply in such a way that guarantees the existence of some $k$-partition $\mathcal{C}^*$ of $V$ such that

$$\text{cost}_{G_t}^{\text{FP}}(\mathcal{C}^*) \leq \ell.$$

The game terminates when there exists a unique $k$-partition $\mathcal{C}^*$ such that $\text{cost}_{G_t}^{\text{FP}}(\mathcal{C}^*) \leq \ell$.

The query complexity of the $\ell$-PL$^{\text{FP}}$ problem can be characterized as

$$\min_{Q \in \mathcal{Q}} \max_{R \in \mathcal{R}} \text{Game}^{\text{FP}}(Q, R),$$

where $\mathrm{Game}^{\mathrm{FP}}(Q, R)$ indicates the number of iterations in the $\mathrm{RUCC}^{\mathrm{FP}}$ game when questioner strategy $Q$ plays against responder $R$. As in Appendix E.1, this characterization allows us to lower bound the query complexity of the problem by analyzing the $\mathrm{RUCC}^{\mathrm{FP}}$ game:

**Lemma 22.** *Let $f$ be a non-negative real valued function. If there exists a responder strategy $R^*$ such that*

$$\forall n, k, \ell \in \mathbb{N}_0 : \quad \min_{Q \in \mathcal{Q}} \mathrm{Game}^{FP}(Q, R^*) \geq f(n, k, \ell)$$

*then the query complexity of learning partitions with $\ell$ false positives is at least $f(n, k, \ell)$.*

The proof is entirely analogous to that of Lemma 20.

In Theorem 25 from Appendix F.2 we show that there exists a responder strategy $R^*$ which satisfies:

$$\min_{Q \in \mathcal{Q}} \mathrm{Game}^{\mathrm{FP}}(Q, R^*) \geq \frac{(\ell + 1)(n - k + 1))}{2} = \Omega(\ell(n - k)).$$

This, together with Lemma 22 immediately implies Theorem 21.

# F    One-sided error lower bounds strategies

## F.1    A responder strategy for the $\mathrm{RUCC}^{\mathrm{FN}}$ game

In this section we show that there exists a responder strategy for the $\mathrm{RUCC}^{\mathrm{FN}}$ game that guarantees the game lasts at least $\Omega(\ell k^2)$ iterations. We consider a responder strategy which we call the $(k + 1)$-*groups responder strategy*, denoted $R_{(k+1)}$. In this strategy, the responder fixes an arbitrary $(k + 1)$-partition of $V$, denoted $\mathcal{C}^* = \{C_i^*\}_{i=1}^{k+1}$. We define $R_{(k+1)}$ as the responder strategy in which the responder replies consistently with $\mathcal{C}^*$ whenever the rules of the $\mathrm{RUCC}^{\mathrm{FN}}$ game allow, i.e. whenever this does not cause the cost of every $k$-partition to exceed $\ell$. We lower bound the number of iterations needed for any questioner strategy to terminate the $\mathrm{RUCC}^{\mathrm{FN}}$ game when playing against $R_{(k+1)}$:

**Theorem 23.** *The $(k + 1)$-groups responder strategy $R_{(k+1)}$ satisfies*

$$\min_{Q \in \mathcal{Q}} \mathrm{Game}^{FN}(Q, R_{(k+1)}) \geq (\ell + 1)\left(\binom{k+1}{2} - 1\right) = \Omega(\ell k^2)$$

*queries.*

*Proof.* For $i, j \in [k + 1]$, consider the $k$-partitions $\mathcal{C}_{ij}$ obtained by starting with $\mathcal{C}^*$ and merging $C_i^*$ and $C_j^*$ i.e.:

$$\mathcal{C}_{ij} = \{C_a^*\}_{a \in [k+1] \setminus \{i,j\}} \cup \{C_i^* \cup C_j^*\}.$$

We show that the questioner cannot increase the cost of more than one of these partitions $\mathcal{C}_{ij}$ on any single iteration (a claim which we prove afterwards):

**Claim 24.** *Suppose that $\mathrm{cost}_{G_t}^{FN}(\mathcal{C}_{ij}) \leq \ell$. Then on iteration $t + 1$ the cost of $\mathcal{C}_{ij}$ increases only if the questionner submits a query $uv$ for $u \in C_i^*$ and $v \in C_j^*$.*

We now prove Theorem 23 using Claim 24. Consider the potential function:

$$\phi^{(t)} := \sum_{\substack{i, j \in [k+1] \\ i \neq j}} \min\{\mathrm{cost}_{G_t}^{\mathrm{FN}}(\mathcal{C}_{ij}), \ell + 1\}. \tag{4}$$

Note that, $\phi^{(0)} = 0$ and, by Claim 24, at every step $t$ of the game we have:

$$\phi^{(t+1)} \leq \phi^{(t)} + 1. \tag{5}$$

Suppose that the game terminates at iteration $T$. Then by the rule of the $\mathrm{RUCC}^{\mathrm{FN}}$ game, for all but one $k$-partitions $\mathcal{C}$, it must be the case that $\mathrm{cost}_{G_T}^{\mathrm{FN}}(\mathcal{C}) > \ell$. In particular, all but at most one of the partitions $\mathcal{C}_{ij}$ must have cost at least $\ell + 1$, implying:

$$\phi^{(T)} \geq (\ell + 1)\left(\binom{k+1}{2} - 1\right), \tag{6}$$

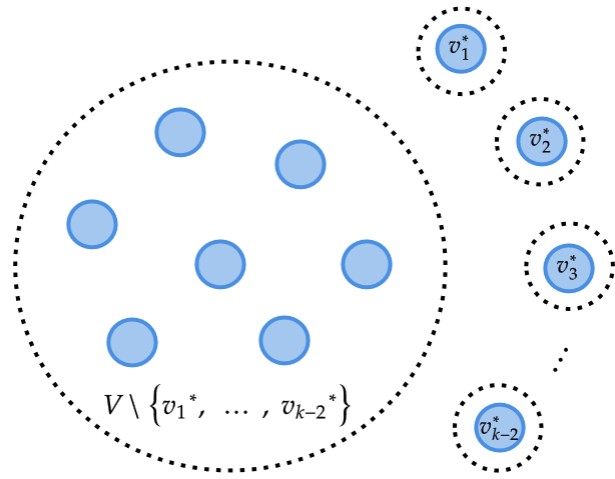

Figure 2: The partition $\mathcal{C}^*$ used in the $(k-1)$ responder strategy.

and hence:

$$T \geq (\ell + 1) \left( \binom{k+1}{2} - 1 \right) = \Omega(\ell k^2),$$

as needed. $\qquad\square$

We conclude this section by proving Claim 24.

**Proof of Claim 24.** Consider $G_t$ such that $\mathrm{cost}^{\mathrm{FN}}_{G_t}(\mathcal{C}_{ij}) \leq \ell$ and such that the RUCC$^{\mathrm{FN}}$ Game has not yet terminated. Assume the query submitted by the questioner is not of the form $uv$ for any $u \in C_i^*$, $v \in C_j^*$. For any such $uv$, by the definition of $\mathcal{C}_{ij}$ the pair $u, v$ is in the same cluster with respect to partition $\mathcal{C}_{ij}$ if and only if the pair is in the same cluster with respect to partition $\mathcal{C}^*$. Thus if the responder is able to return an answer consistent with $\mathcal{C}^*$, this response will also be consistent with $\mathcal{C}_{ij}$, and as a result $\mathrm{cost}^{\mathrm{FN}}_{G_t}(\mathcal{C}_{ij}) = \mathrm{cost}^{\mathrm{FN}}_{G_{t+1}}(\mathcal{C}_{ij})$.

It remains to show that the responder is able to reply consistently with $\mathcal{C}^*$ on such a query. By assumption $\mathrm{cost}^{\mathrm{FN}}_{G_t}(\mathcal{C}_{ij}) \leq \ell$, and as shown above responding consistently to the query $uv$ in a manner consistent with $\mathcal{C}^*$ will ensure $\mathrm{cost}^{\mathrm{FN}}_{G_t}(\mathcal{C}_{ij}) = \mathrm{cost}^{\mathrm{FN}}_{G_{t+1}}(\mathcal{C}_{ij}) \leq \ell$. Thus if the responder replies to query $uv$ consistently with $\mathcal{C}^*$, there will still exist a $k$-partition–namely, $\mathcal{C}_{ij}$–such that $\mathrm{cost}^{\mathrm{FN}}_{G_{t+1}}(\mathcal{C}_{ij}) \leq \ell$. Thus the rules of the RUCC$^{\mathrm{FN}}$ game allow the responder to reply to query $uv$ consistently with $\mathcal{C}^*$, and by definition of strategy $R_{(k+1)}$ the responder will do so. Thus the cost of $\mathcal{C}_{ij}$ at iteration $(t+1)$ does not increase when the questioner plays such a pair $uv$. $\qquad\square$

## F.2 A responder strategy for the RUCC$^{\mathrm{FP}}$

In this section, we show that there exists a responder strategy for the RUCC$^{\mathrm{FP}}$ game that guarantees that the game lasts at least $\Omega(\ell(n-k))$ iterations. In particular, we consider a responder strategy which we call the $(k-1)$-*groups responder strategy*, denoted $R_{(k-1)}$. This strategy fixes $k-2$ arbitrary elements $v_1^*, ..., v_{k-2}^*$ and then fixes the following $(k-1)$-partition:

$$\mathcal{C}^* := \{V \setminus \{v_1^*, ..., v_{k-2}^*\}, \{v_1^*\}, ..., \{v_{k-2}^*\}\}.$$

(See Figure Figure 2). Note that when $k = 2$, the set of $\{v_i^*\}$ is just empty and the partition $\mathcal{C}^*$ consists of a single set equal to $V$. We define the $(k-1)$-groups strategy, as the responder strategy in which the responder replies consistently with $\mathcal{C}^*$ whenever the rules of the RUCC$^{\mathrm{FP}}$ game allow them to.

**Theorem 25.** *The $(k-1)$-groups responder strategy $R_{(k-1)}$ satisfies:*

$$\min_{Q \in \mathcal{Q}} \text{Game}^{FP}(Q, R_{(k-1)}) \geq \frac{(\ell+1)(n-k+1))}{2} = \Omega(\ell(n-k)).$$

*Proof.* We will prove the theorem by considering a subset of the partitions which are hard for the questioner to differentiate among when playing against this responder strategy. Let $S^* := V \setminus \{v_1^*, ..., v_{k-2}^*\}$, and for any $v \in S^*$ let $\mathcal{C}_v$ be the partition:

$$\mathcal{C}_v := \{\{v\}, S^* \setminus \{v\}, \{v_1^*\}, ..., \{v_{k-2}^*\}\}.$$

By definition of the RUCC$^{\text{FP}}$ game, at the end of the final iteration $T$, we must have $\text{cost}_{G_T}^{\text{FP}}(\mathcal{C}_v) > \ell$ for all but one of the partitions $\{\mathcal{C}_v\}_{v \in S^*}$. In particular, consider the potential function:

$$\phi^{(t)} \stackrel{\text{def}}{=} \sum_{v \in S^*} \min\{\text{cost}_{G_t}^{\text{FP}}(\mathcal{C}_v), \ell+1\}.$$

Then, if the game ends at iteration $T$, we must have:

$$\phi^{(T)} \geq (\ell+1)(|S^*|-1) = (\ell+1)(n-k+1). \tag{7}$$

On the other hand, we have:

$$\phi^{(0)} = 0. \tag{8}$$

We show that this potential $\phi^{(t)}$ changes by a small amount on any given iteration of the RUCC$^{\text{FP}}$ game:

**Claim 26.** *If $\phi^{(t)} < (n-k+1)(\ell+1)$, then $\phi^{(t+1)} \leq \phi^{(t)} + 2$.*

An immediate consequence of Claim 26 and Equations (7) and (8) is that the number of iterations required for the game to terminate is at least $T \geq (n-k+1)(\ell+1)/2$, yielding the statement of Theorem 25. $\qquad\square$

We conclude by proving Claim 26.

**Proof of Claim 26.** Let $uv$ be the query submitted by the questioner on iteration $t+1$. We will split the proof of the theorem into cases.

**Case 1** Suppose $\{u, v\} \not\subseteq S^*$. Then, if the responder answers in a way that's consistent with $\mathcal{C}^*$, their answer will also be consistent with $\mathcal{C}_w$ for every $w \in S^*$. By consequence of the assumption that $\phi^{(t)} < (n-k+1)(\ell+1)$, a simple counting argument implies that there exists an element $w_t \in S^*$ such that:

$$\text{cost}_{G_t}^{\text{FP}}(\mathcal{C}_{w_t}) \leq \ell.$$

Since $w_t$ as defined above satisfies $\text{cost}_{G_t}^{\text{FP}}(\mathcal{C}_{w_t}) \leq \ell$, this implies that the responder can answer in a way consistent with $\mathcal{C}^*$ without violating the rules of the game. By the definition of the strategy, the responder then will answer in a way consistent with $\mathcal{C}^*$ (and with all partitions $\mathcal{C}_w$) and hence, for all $w \in S^*$: $\text{cost}_{G_{t+1}}^{\text{FP}}(\mathcal{C}_w) = \text{cost}_{G_t}^{\text{FP}}(\mathcal{C}_w)$, giving $\phi^{(t+1)} = \phi^{(t)}$.

**Case 2.a** Suppose that $\{u, v\} \subseteq S^*$, and that the responder replies in a way that's consistent with $\mathcal{C}^*$. In this scenario, for any $w \in S^* \setminus \{u, v\}$ we have $\text{cost}_{G_{t+1}}^{\text{FP}}(\mathcal{C}_w) = \text{cost}_{G_t}^{\text{FP}}(\mathcal{C}_w)$, while $\text{cost}_{G_{t+1}}^{\text{FP}}(\mathcal{C}_u) \leq \text{cost}_{G_t}^{\text{FP}}(\mathcal{C}_u) + 1$ and $\text{cost}_{G_{t+1}}^{\text{FP}}(\mathcal{C}_v) \leq \text{cost}_{G_t}^{\text{FP}}(\mathcal{C}_v) + 1$. This immediately implies $\phi^{(t+1)} \leq \phi^{(t)} + 2$.

**Case 2.b** Suppose that $\{u, v\} \subseteq S^*$, and that the responder replies in a way that is **not** consistent with $\mathcal{C}^*$, i.e. the responder replies to the query $uv$ with $-1$. Then by the definition of the responder strategy, it must be the case that answering $+1$ to the query would violate the rules of the game, i.e. it would cause every $k$-partition $\mathcal{C}$ of $V$ to have $\text{cost}_{G_{t+1}}^{\text{FP}}(\mathcal{C}) \geq \ell+1$. But, for every $w \in S^* \setminus \{u, v\}$, the cost of the partition $\mathcal{C}_w$ does not increase if the response

to the query $\{u, v\}$ is $+1$. Hence, it must be the case that $\mathrm{cost}^{\mathrm{FP}}_{G_t}(\mathcal{C}_w) \geq \ell + 1$ for all such $w$. Hence, for any $w \in S^* \setminus \{u, v\}$:

$$\min\{\ell + 1, \mathrm{cost}^{\mathrm{FP}}_{G_{t+1}}(\mathcal{C}_w)\} = \ell + 1 = \min\{\ell + 1, \mathrm{cost}^{\mathrm{FP}}_{G_t}(\mathcal{C}_w)\}. \tag{9}$$

On the other hand, since the responder is replying to the query $uv$ with $-1$, their answer is simultaneously consistent with both $\mathcal{C}_u$ and $\mathcal{C}_v$. We then have:

$$\mathrm{cost}^{\mathrm{FP}}_{G_{t+1}}(\mathcal{C}_u) = \mathrm{cost}^{\mathrm{FP}}_{G_t}(\mathcal{C}_u) \text{ and } \mathrm{cost}^{\mathrm{FP}}_{G_{t+1}}(\mathcal{C}_v) = \mathrm{cost}^{\mathrm{FP}}_{G_t}(\mathcal{C}_v). \tag{10}$$

In particular, Equations (9) and (10) directly imply $\phi^{(t+1)} = \phi^{(t)}$.

This concludes the proof of Claim 26 (and in turn the proof of Theorem 25).

$\square$

