# OpenReview forum: "Optimal Algorithms for Learning Partitions with Faulty Oracles"
_NeurIPS.cc/2024/Conference — NeurIPS 2024 poster_

### Official Review · Reviewer_2QcX · 2024-06-21

**Soundness:** 4
**Presentation:** 4
**Contribution:** 3
**Rating:** 8
**Confidence:** 4

**Summary:**

This paper studies the problem to recover an exact $k$ partition of a set with access to a same-cluster oracle that is allowed to lie $\ell$ times. This papers gives an algorithm with optimal query complexity up to constants and a lower bound.

**Strengths:**

1. The result of this paper is clean and complete. The algorithm's query complexity matches the lower bound up to constants.

2. The main algorithm is concise, simple and elegant. The idea of the algorithm captures the problem well and has proved optimal guarantees.

3. The lower bound is non-trivial and has some interesting ideas.

4. This paper is very well-written. The notations and explanations are clear. I didn't even catch a single typo. Enough background and motivation is included in the paper. The paper is cohesive and organized, easy to follow. Math and algorithmic ideas are explained clearly.

5. I think this paper should be a spotlight.

**Weaknesses:**

I'm very satisfied with this paper, just two things I think it can improve on the writing.

1. The algorithm is quite simple and intuitive. On the other hand, the lower bound is more complicated and more technical. I think it might be better to write less on the algorithm but explain more on the lower bound, especially how to construct a good responder's strategy.

2. I think it's worth mentioning what's the optimal algorithm for the no-error oracle and compare your algorithm with theirs.

Also something I would not like to call it a weakness since I think it's beyond the scope of the paper:

1.  You justified the inconsistent error assumption (but I think the consistent error assumption can also be justified). But from the pure theoretical point of view, this assumption does make the algorithm design a lot easier and less interesting since the algorithm can make the same query many times. If the error model prevents such behavior of the algorithm, it is more interesting.

**Questions:**

1. Is it possible to get better query complexity bounds if the goal is to recover the partition approximately but not exactly?

2. I mentioned consistent error above. I'm also thinking a more generalized adversarial error where it could be stochastic, for example, the expected number of error is $\ell$. Does stochasticity makes things harder?

**Limitations:**

The limitation of the paper is well addressed.

---

> ### Author Rebuttal · Authors · 2024-08-02
>
> We thank you for your thorough review.
>
> We address your questions and comments in order. Regarding the writing: we agree that the details of the lower bound proof are technically more challenging and arguably more mathematically interesting, but we had previously decided to prioritize the algorithm since we thought the community might value it more. We will be happy to write more about the intuition behind the design of the responder strategy in the main body of the paper, and make the relevant appendix more detailed if the paper gets accepted.
>
> In terms of comparing with the error-free algorithm, there certainly are similarities between the two algorithms, and we would be happy to include an extra appendix elaborating on this comparison. In fact, the algorithms proposed in this work nearly-generalize the original algorithms for the error-free regime; when $k$ is known, setting $\ell_{{yes}}=\ell_{{no}}=0$ causes the algorithm to perform a nearly-equivalent procedure to the corresponding $k$-known algorithm of Reyzin and Srivastava, and when $k$ is unknown, this setting yields an algorithm that is exactly equivalent.
>
> Regarding the inconsistent error assumption: while it is true that this assumption guarantees that the problem is solvable given unlimited queries, and makes the problem easier for the algorithm, we believe that the value of our result lies in understanding the query complexity in this setup.
>
> If the model is chosen to prevent the algorithm from querying the same pair multiple times, then exact recovery is impossible even for very small values of $\ell$ (say $\ell = 3$). This follows from a previous result of Reyzin et al. This naturally leads to your question of partial recovery. This is an interesting research direction, and we believe that answering it may require a non-trivial amount of further work. The same applies to the stochastic error setting you suggested. It is worth noting that if one knows the expected number of errors, a simple application of Markov's inequality would allow one to run our algorithm with settings that guarantee success with probability 1-$\delta$ while making a number of queries that grows linearly in $1/\delta$. Stronger statements could likely be made by making further assumptions about the distribution of $\ell$. We are hopeful that our work will serve as a first step towards answering more questions in this problem space.
>
> We thank you again for your time and your help in improving the quality of our paper.
>
> Reyzin et al. Learning and Verifying Graphs using Queries with a Focus on Edge Counting

---

> > ### Comment · Reviewer_2QcX · 2024-08-09
> >
> > Thank you for your response! I hope to see this paper at NeurIPS 2024!

---

### Official Review · Reviewer_GCFN · 2024-06-28

**Soundness:** 4
**Presentation:** 3
**Contribution:** 3
**Rating:** 6
**Confidence:** 4

**Summary:**

**[Setting]**:
This paper studies the problem of clustering n items into k clusters using an oracle that adversarially answers same-cluster queries for item pairs under the constraint that it makes at most $\ell$ errors for a known constant $\ell$. The goal is to exactly recover all clusters always (instead of just w.h.p.).

**[Contributions]**:
1. A lower bound on the number of queries when k is known/unknown. The authors formulate the problem in terms of Chip-Liar game to get this result.
2. For known k, an algorithm that iteratively merges cluster using two heuristics:
    1. If there is a (k+1)-clique of all -1s then the oracle has returned at least one false negative
    2. More than $\ell$ "+1" responses from oracle for a given pair guarantees that it is in the same cluster
3. Sample complexity of the proposed algorithm that matches the lower bound.

The results extend to a more general problem where individual limits on false positive and false negative errors are known.

The paper also studies an algorithm for k-unknown case in the appendix. Sample complexity in this case is not optimal.

**Strengths:**

1. The problem of guaranteed exact cluster recovery in the presence of noise is new. Having a hard limit on the number of errors made by the oracle makes this possible. Given concrete applications, this would be an interesting direction to explore.
2. The sample complexity of the proposed algorithm matches the derived lower bound when k is known.
3. The connection to Chip-Liar game for deriving the lower bound is interesting.

**Weaknesses:**

1. The problem setting (oracle making at most $\ell$ errors with $\ell$ being a known constant) is not very practical in my opinion, which in-turn makes it hard to judge the significance of the results. Even for the examples given in the paper (L23-32), it is not clear why the oracle will make at most $\ell$ errors (e.g., an experiment failing in bioinformatics) or why $\ell$ will be known in advance. Do the authors have concrete applications in mind?
2. Clarity-wise, while the details in the paper are mostly clear, it would be helpful to include more details from the appendix into the main paper. For example, the following can be included by making Section 3 more concise,
    1. What does "The position of a chip on the board will then be equal to the cost of the corresponding partition .." (L234-235) mean?
    2. Some high-level details about the unknown-k algorithm.
    3. Some intuition about why false-negative and false-positive error budgets inherently have a different contribution towards minimum sample complexity

**Questions:**

Please respond to point 1 under weakness


**Minor suggestions**:
1. Typo in L100 - "A many" -> "Many"
2. A more recent paper (Gupta et al. 2024) studies a more general setting than Chen et al. (2023), which is closest to your work.


Gupta et al. Clustering Items From Adaptively Collected Inconsistent Feedback - AISTATS, 2024

**Limitations:**

Yes

---

> ### Author Rebuttal · Authors · 2024-08-05
>
> We thank you for taking the time to provide thorough feedback on our work. We begin by addressing your main question. After that, we discuss some of the weaknesses you mentioned.
>
> **"... it is not clear why the oracle will make at most $\ell$ errors ... or why $\ell$ will be known in advance. Do the authors have concrete applications in mind?"**
>
> The results in the paper characterize the minimum query cost needed in order to achieve a desired level of robustness to a number of errors. In particular, one may think of $\ell$ as not being given or known, but rather as being chosen by the user based on their desired error tolerance. This was briefly described in the introduction (L55-56). In the "global" response to all authors we discuss two more examples in which our model may be applied, which we will include in the paper if it gets accepted. The first of these (Example 1: Robustness to Misinformation) describes a situation in which the number of corruptions that will occur is finite and does not scale with the number of queries submitted, and in this example the learner may set the value of $\ell$ depending on their desired robustness to adversarial error. This type of cost-robustness trade-off is common in fields like information and coding theory, in which one is interested in the best achievable rate of communication subject to some fixed worst-case error.
>
> Dually, one may also think of our results as resolving the optimal error robustness subject to a constraint on the number of queries. Our second example (Example 2: Trustworthy Science) outlines a scenario in which one may benefit from understanding this trade-off.
>
> We note that it is impossible to design a protocol that allows the learner to exactly recover the full partition while remaining robust to a number of errors that grows as a constant fraction of the queries made. This follows trivially from known impossibility results (e.g. Proposition 1 of Reyzin et al.), and we will add a precise statement to the paper if it gets accepted.
>
> We briefly comment that in the $k$-known setting, it is actually not necessary for the number of false negatives ($\ell_{no}$) to be bounded or known in advance in order to run our algorithm and achieve optimal query complexity. Rather the value of $\ell_{no}$ that appears in the query complexity analysis will be the true number of false-negative responses that occur during the execution of the algorithm. In particular, this implies that in the version of the problem in which no false positives can occur, one does not need to set any upper bound on the error at all. Admittedly this may not be apparent since we pass $\ell_{no}$ as one of the arguments in the pseudocode, so we will remove it for clarity. In contrast, in the $k$-unknown setting the algorithm does require a user-chosen value for $\ell_{no}$. Here, a simple argument shows that it is impossible to guarantee one has found the correct answer unless they can upper bound both $\ell_{yes}$ and $\ell_{no}$.
>
> Effectively conveying the specifics of the error model is a key aspect of communicating our results, and we thank you for prompting us to clarify the role of $\ell$. We will update the manuscript to increase the emphasis on the above discussion, and we believe this will strengthen our paper.
>
> **Further Discussion of Weaknesses**
>
> **``...it would be helpful to include more details from the appendix into the main paper.''** We thank you for this suggestion and we agree. If the paper is accepted, we will include an overview of the $k$-unknown algorithm, discuss intuition for the asymmetry between false-negative and false-positive errors, and elaborate on the role of the chip board analogy.
>
> We now briefly answer the question: **``What does 'The position of a chip on the board will then be equal to the cost of the corresponding partition ...' (L234-235) mean?''** In the $\ell$-PL constrained version of the Chip-Liar game, each chip corresponds to one partition of $n$ elements into $k$ groups. When the questioner submits a query ("Are $u$ and $v$ in the same group?"), the responder answers either "yes" or "no". All chips whose partitions are inconsistent with the response (i.e. if the responder answers "yes", then all partitions where $u$ and $v$ are not in the same group) advance by one position on the board. In L234-235 we highlight that under this procedure, the position of the chip on the board is then equal to the cost of the partition as a feasible solution to an instance of a correlation clustering problem.
>
> Regarding the **"Minor Suggestions"** we were not aware of the recent work of Gupta et al. and we appreciate the pointer. We will make sure to discuss and cite it appropriately. We will also fix the typo you found.
>
> We conclude with a quick remark in response to the following: **"The paper also studies an algorithm for $k$-unknown case in the appendix. Sample complexity in this case is not optimal."** While it is true that, in the $k$-unknown setting, our algorithm does not achieve the optimal query complexity in **every** variant of the problem, it does achieve the optimal query complexity in the (unweighted) $\ell$-PL problem, i.e. the main variant in which false positive and false negative responses are penalized equally.
>
> Lastly, we wish to thank you again for your time and your help in improving the paper. We hope that in light of the clarifications on the error model you may deem our contributions more substantial.
>
> Reyzin et al. Learning and Verifying Graphs using Queries with a Focus on Edge Counting

---

> > ### Comment · Reviewer_GCFN · 2024-08-12
> >
> > Thank you for responding to my questions. I very much appreciate the added motivation and applications. Consequently, I have increased my score from 4 to 6. All the best for your submission ! :)

---

### Official Review · Reviewer_BYtw · 2024-07-08

**Soundness:** 3
**Presentation:** 3
**Contribution:** 3
**Rating:** 6
**Confidence:** 4

**Summary:**

The paper studies the problem of finding a hidden partition into $k$ clusters of a given universe.
In many applications an algorithm has only access to a same-cluster oracle. A query to this oracle reveals whether two elements belong to the same cluster or not. This problem has been previously studied and tight bounds on the query complexity, i.e., the minimal number of queries required to solve the problem, are known (Reyzin and Srivastava, and Liu and Mukherjee).
In this paper, the authors add the realistic assumption that the same-cluster oracle may not always reveal the correct answer. In their model, they (in advance) set a number \ell which bounds the maximum number of wrong answers which the oracle is allowed to make. The goal of an algorithm is still to compute the hidden cluster with as few queries as possible. In particular, for the same tuple of elements the oracle may give different answers for different oracle calls, and the algorithm does not receive any information on whether the response of the oracle was correct or not. The authors present an algorithm and analyze its query complexity. This bound is generally larger than in the setting with a correct oracle, and depends on the parameter \ell. If \ell=0, the presented analysis recovers the results by Reyzin and Srivastava. Furthermore, they give a tight lower bound using an argumentation based on Renyi-Ulam games and correlation clustering.
They moreover study a slightly more general setting where the algorithm can set in advance more fine-grained bounds on how many false positive and false negative answers the oracle can give, and, for all problems they consider both, the setting where the number of hidden clusters k is known or not.

**Strengths:**

- I think that the problem is important and appreciated by the ML-community, as clustering is a fundamental problem in machine learning. Moreover, the assumption that a same-cluster oracle may not always be correct seems quite reasonable and realistic. Thus, I think that this problem and the presented results could have many applications and an impact in certain areas.
- The authors give a tight analysis of the considered algorithms.
- Despite being tight up to constants, the main algorithm is well-presented, and easy to understand and implement.
- Overall, I think that the paper is well-written and seems technically sound.

**Weaknesses:**

- I think the main weakness of the model is that the upper bound \ell on the number of faulty oracle responses must be set in advance and stays fixed. This could be a major drawback when applying this model and the algorithm in practice, because it seems not clear why a faulty oracle should be consistent with such a bound.
- It seems that the main algorithms is quite similar to the algorithm without faulty oracle. I think it would be helpful for the reader to have a paragraph where the difference to this original algorithm is explained.

Further comments:
- Line 236: missing 'and'

**Questions:**

Is there anything known for the setting where the number \ell is unknown to the algorithm, and it only appears in the analysis? I.e. a strong lower bound or an obvious workaround? Such insights or discussions could make the main weakness less severe.

**Limitations:**

Limitations have been addressed.

---

> ### Author Rebuttal · Authors · 2024-08-05
>
> We thank you for your response.
>
> We begin by answering your question: **"Is there anything known for the setting where the number $\ell$ is unknown to the algorithm, and it only appears in the analysis?"**. In fact in the $k$-known setting, the algorithm presented in the paper does not require knowledge of $\ell_{no}$ in order to be correct. Admittedly this may not be apparent since we pass $\ell_{no}$ as one of the arguments in the pseudocode, so we will make sure to remove it in order to make this less confusing to future readers.
>
> A consequence of this is that in the $k$-known setting, one does not require any a priori knowledge of $\ell_{no}$, nor are they required to set any upper bound for $\ell_{no}$ in advance, in order to achieve the optimal query complexity both in the weighted $\ell$-PL problem and in the version of the problem in which only false negatives can occur. In particular, in this latter version, the learner would not require any knowledge of the error at all in order to achieve the optimal query complexity.
>
> Whether one could design an algorithm that also requires no knowledge of $\ell_{yes}$ in the $k$-known setting is an interesting question and we will make sure to discuss it in the "Conclusions, Limitations, and Future Directions" section of the paper.
>
> We note that in the $k$-unknown setting, without knowledge of $\ell_{no}$ and $\ell_{yes}$ it is impossible for an algorithm to make a finite number of queries and then output a partition that is guaranteed to be correct. To see this, suppose the oracle answers in a way that is consistent with some fixed partition $\mathcal{C}$, then (1) if $\ell_{no}$ is unknown to the learner it is impossible for them to establish whether $\mathcal{C}$ is the true partition or a refinement of it, on the other hand, (2) if $\ell_{yes}$ is unknown to the learner then it is impossible for them to tell whether $\mathcal{C}$ is the true partition or a coarsening of it.
>
> If the paper is accepted, we will add a summary of the above discussion to the manuscript. We will also be happy to elaborate on the similarities and differences with the algorithm for the error-free version of the problem. Finally, we have fixed the typo you found.
>
> We thank you again for your time and your useful comments.

---

> > ### Comment · Reviewer_BYtw · 2024-08-10
> > **Response to Rebuttal**
> >
> > I thank the authors for their rebuttal.
> >
> > As far as I undernstand, there is still a major limitation to the result (at least for the k-known setting) because an algorithm requires to know $\ell$ upfront (expect that in the weighted setting $\ell_{no}$ can be unknown), which I think is not very realistic. I see that, in the k-unknown setting, there is a strong theorical lower bound, and I appreciate that you provided it. Overall, my opinion about the paper and my score remains: The paper has some nice insights and results, but also a major limitation to the model.

---

### Official Review · Reviewer_E2Nr · 2024-07-08

**Soundness:** 3
**Presentation:** 3
**Contribution:** 3
**Rating:** 6
**Confidence:** 4

**Summary:**

This paper studies the query complexity of clustering with a faulty oracle. Given a set of $n$ points $V$, which is partitioned into $k$ hidden clusters, the learner wants to recover the hidden partition by querying whether two points are in the same clusters or not. There has been a line of work that studies the query complexity of the problem where the response of each query has iid error. This paper studies a different query model, where the learner is allowed to make repeat queries for the same pair of points but the responses could be adversarially flipped at most $\ell$ times. This paper provides lower bounds for the query complexity of several variants of the problem and also designs efficient learning algorithms with a query complexity matching the lower bound.

**Strengths:**

1. The paper establishes a novel relation between the clustering problem and the Rényi-Ulam liar games, which could potentially be useful for proving lower bounds for other learning problems.
2. The algorithm designed in this paper involves non-trivial techniques and has a query complexity that matches the lower bound proved in the paper.

**Weaknesses:**

My main concern is about the significance of the learning model studied in the paper.
For graph clustering problems, the error is usually defined over the graph instead of over the queries, and sometimes repeated queries are not allowed. This is because sometimes by allowing the use of repeated queries, the learning problem could be easy to solve. For example, when iid noise is presented.
In this paper, the error is defined over an unbounded sequence of queries but only allows a constant number of mistakes to happen. In particular, knowing the number of mistakes seems to be very important to make the learning algorithm designed in this paper work. These two points seem to be too idealized to model problems that arise from real applications.

**Questions:**

My questions are about the weakness pointed out above.
1. Can you provide any real applications that motivated the study of such a learning model? (Only a constant number of mistakes are made over the queries and such a number is known)
2. How would the learner know the error parameter $\ell$ in advance and if we do not have the parameter $\ell$ as input would it be possible to achieve exact recovery?
3. If repeated queries are not allowed and the mistakes are placed by an adversary, would it still be possible to (almost) recover the underlying clusters?

---

> ### Author Rebuttal · Authors · 2024-08-02
>
> We thank you for your helpful comments and questions.
>
> We begin by answering your questions, and we then discuss some of the weaknesses that were raised.
>
> **1. "Can you provide any real applications that motivated the study of such a learning model? (Only a constant number of mistakes are made over the queries and such a number is known)"**
>
> In the "global" response to all reviewers, we provide further examples. The first example we provide (Example 1: Robustness to Misinformation) gives a setting in which it is reasonable to assume that the user can estimate an upper bound on the number of errors, and this quantity does not grow with the number of queries made.
>
> In general the aim of this project is to understand the fundamental trade-off between robustness to errors and query complexity when making same-cluster queries. In this context, the $\ell$-faulty oracle may not necessarily model the behavior of a system, but rather serves as an analytical tool to prove formal guarantees about this tradeoff. The second example (Example 2: Trustworthy Science) describes a setting in which it may helpful to understand the trade-off between robustness to error and number of queries independently of assumptions about the error model.
>
> **2. "How would the learner know the error parameter $\ell$ in advance and if we do not have the parameter $\ell$ as input would it be possible to achieve exact recovery?"**
>
> One should think of $\ell$ not as being given or known, but rather as being chosen by the user based on their desired error tolerance. This was briefly described in the introduction (L55-56). We will be happy to further emphasize this perspective in the paper. For the full recovery guarantees to apply, the parameter $\ell$ does not need to be equal to the number of errors occurring, but it could simply be an upper bound to that quantity.
>
> We briefly comment that in the $k$-known setting, it is actually not necessary for the number of false negatives ($\ell_{no}$) to be bounded or known in advance in order to run our algorithm and achieve optimal query complexity. Rather the value of $\ell_{no}$ that appears in the query complexity analysis will be the true number of false-negative responses that occur during the execution of the algorithm. In particular, this implies that in the version of the problem in which no false positives can occur, one does not need to set any upper bound on the error at all. Admittedly this may not be apparent since we pass $\ell_{no}$ as one of the arguments in the pseudocode, so we will remove it for clarity. In contrast, in the $k$-unknown setting the algorithm does require a user-chosen value for $\ell_{no}$. Here, a simple argument shows that it is impossible to guarantee one has found the correct answer unless they can upper bound both $\ell_{yes}$ and $\ell_{no}$.
>
>
> **3. "If repeated queries are not allowed and the mistakes are placed by an adversary, would it still be possible to (almost) recover the underlying clusters?"**
>
> Approximate recovery of the partition in this setting is an interesting research direction and remains an area for future work. We note that if repeated queries are not allowed, exact recovery becomes impossible even for small values of $\ell$ ($\ell=3$). This follows from a simple extension of a previous result of Reyzin et al. and we would be happy to highlight this fact in the paper if accepted. These impossibility results are in part what motivated the study of our paper's model in the first place.
>
> **Discussion of weaknesses.**
> While it is true that allowing repeated queries makes the task easier for the learner, one should note that assuming that the errors are made over the graph--as opposed to over the queries--may also simplify the problem, since typically one assumes that an adversary would commit to corruptions before observing the set of queries being made. These graph-based error models may not be suited to modeling every setting. Our first example (Example 1: Robustness to Misinformation) emphasizes why one may want to model a stronger adversarial behavior in this sense.
>
> With regards to the comment "knowing the number of mistakes seems to be very important to make the learning algorithm designed in this paper work," we emphasize that alongside providing an algorithm, a core goal of this paper is characterizing the tradeoff between error occurrence and query complexity. The cost-robustness tradeoff studied in this work is common in fields like information and coding theory, in which one is interested in the best achievable rate of communication subject to some fixed worst-case error.
>
>
> We thank you again for your helpful comments, and we hope that, in light of this discussion, you may deem our contributions more substantial. We are happy to answer any further questions during the discussion period.
>
>
> Reyzin et al. Learning and Verifying Graphs using Queries with a Focus on Edge Counting

---

> > ### Comment · Reviewer_E2Nr · 2024-08-09
> >
> > Thanks for spending time making the response. I would like to keep my score.

---

### Author Rebuttal · Authors · 2024-08-05

We thank all the reviewers for their time. Multiple reviewers have asked for more examples in which our model could be applied. Below, we give two more general motivating examples, in the tech and scientific domains respectively, illustrating the role of $\ell$ in learning tasks.

**Example 1: Robustness to Misinformation**

Consider a setting where a user is trying to cluster a dataset by crowdsourcing information in the form of same-cluster questions. However, the user suspects that an ill-intentioned competitor organization is attempting to corrupt the learning process by entering a number of bad actors in the crowd to strategically mislabel queries. If the user selects a new person every time they submit a query, then the number of adversarial answers they encounter is finite and does not grow with the number of queries submitted.

In this scenario, $\ell$ plays the role of a security parameter, and the algorithm is guaranteed to be robust to up to $\ell$-many poisoned responses. The user can set $\ell$ based on, e.g., their prior belief about the resources of the competitor organization. Our results can be interpreted as quantifying the cost (in queries / crowd size) of implementing a fixed security parameter $\ell$.

**Example 2: Trustworthy Science**

Consider a setting in which a scientist is attempting to group items into classes by running experiments that reveal whether two items are in the same class. The scientist has limited resources (e.g. limited materials or time) and can only conduct a finite number of experiments. Our results allow the scientist to derive the maximum number of errors to which their learning procedure can be tolerant, given their fixed query budget. They can use this maximum value as the setting for $\ell$, and then use our algorithms to guide their choice of experiments. Our analysis would then allow them to measure the significance of their findings by quantifying the number of experiments that would need to have failed for the finding to be incorrect.

---

> ### Comment · Area_Chair_BFiL · 2024-08-14
>
> Dear Authors,
> Thanks for all the clarifications you have provided so far. The paper is an enjoyable read. Is it possible to do a comparison with Mazumdar and Saha ([31] in the paper) to show the "adversarial error" query complexity is different from "random error" query complexity? Also the persistent query model is strictly harder to handle than the non-persistent model in the case of random independent errors. Is that the case for (your) adversarial error model? The reference to practitioners here is quite misleading in this context.
>
> I general there is a quite substantial literature on same-cluster queries (eg. multiple papers by Mazumdar and Saha, Chien et al) - it would be useful to include a discussion on this literature.
>
> AC

---

### Decision · Program_Chairs · 2024-09-25

**Decision:**

Accept (poster)

**Comment:**

The reviewers are generally positive about this paper. The query complexity of finding a partition with faulty oracle is a well-motivated problem. However, there is concern regarding how the results are presented and the thoroughness of literature review. In fact, the persistent query model is strictly harder to handle than the non-persistent model in the case of random independent errors. The authors should elaborate if that remains true for the case for the adversarial error model. The reference to practitioners here is quite misleading in this context. Is it possible to do a comparison with Mazumdar and Saha ([31] in the paper) to show the "adversarial error" query complexity is different from "random error" query complexity? In general there is a quite substantial literature on same-cluster queries (eg. multiple papers by Mazumdar and Saha, Chien et al) - it would be useful to include a discussion on and comparison to this literature.